# Achieving health-oriented air pollution control requires integrating unequal toxicities of industrial particles

Di Wu[1,9], Haotian Zheng[2,3,9], Qing Li [1,4] ✉, Shuxiao Wang [2,3] ✉, Bin Zhao [2,3], Ling Jin [5,6], Rui Lyu[7], Shengyue Li[2], Yuzhe Liu[1], Xiu Chen[1], Fenfen Zhang[2], Qingru Wu [2,3], Tonghao Liu[8], Jingkun Jiang[2,3], Lin Wang[1], Xiangdong Li[5], Jianmin Chen [1,4] & Jiming Hao[2,3]

Protecting human health from fine particulate matter (PM) pollution is the ambitious goal of clean air actions, but current control strategies largely ignore the role of source-specific PM toxicity. Here, we proposed health-oriented control strategies by integrating the unequal toxic potencies of the most polluting industrial PMs. Iron and steel industry (ISI)-emitted $PM_{2.5}$ exhibit about one order of magnitude higher toxic potency than those of cement and power industries. Compared with the current mass-based control strategy (prioritizing implementation of ultralow emission standards in the power sector), the proposed health-oriented control strategy (priority control of the ISI sector) could generate 5.4 times higher reduction in population-weighted toxic potency-adjusted $PM_{2.5}$ exposure among polluting industries in China. Furthermore, the marginal abatement cost per unit of toxic potency-adjusted mass of ISI-emitted $PM_{2.5}$ is only a quarter of that of the other two sectors under ultralow emission scenarios. We highlight that a health-oriented air pollution control strategy is urgently required to achieve cost-effective reductions in particulate exposure risks.

Air pollution attributable to particulate matter (PM) is one of the greatest environmental health risks worldwide[1–5]. Industrial sectors, constituting one of the major anthropogenic sources of ambient PM, contribute to approximately 0.9 million premature deaths and are an important global health concern[6,7]. Clean air actions have been adopted earlier in high-income economies (economies with a gross national income per capita of US$13,845 or more in 2022[8], such as the United States and the European Union) to address industrial sources of air pollution[9,10]. Based on total PM mass emissions, these regulations initially focused on the most PM mass emission sectors, including power plants and coal-fired industrial facilities, and then spread to other industrial sectors since the early 1970s[9,11]. Regulations have become increasingly strict to meet the ever-stringent air quality standards over the last four decades[9,12,13]. Referring to the total PM mass control strategies implemented in high income regions, similar emission control policies have been promulgated and prioritized among power plants and other industrial sectors in low- and medium-income regions to combat the unprecedented air pollution over the last two

[1]Department of Environmental Science and Engineering, Shanghai Key Laboratory of Atmospheric Particle Pollution and Prevention, Fudan University, Shanghai 200433, China. [2]State Key Joint Laboratory of Environment Simulation and Pollution Control, School of Environment, Tsinghua University, Beijing 100084, China. [3]State Environmental Protection Key Laboratory of Sources and Control of Air Pollution Complex, Beijing 100084, China. [4]Shanghai Institute of Eco-Chongming (SIEC), 20 Cuiniao Road, Chenjia Town, Chongming District, Shanghai 202162, China. [5]Department of Civil and Environmental Engineering, The Hong Kong Polytechnic University, Hong Kong, China. [6]Department of Health Technology and Informatics, The Hong Kong Polytechnic University, Hong Kong, China. [7]China Huaneng Clean Energy Research Institute, Beijing 102209, China. [8]China National Environmental Monitoring Center, Beijing 100012, China. [9]These authors contributed equally: Di Wu, Haotian Zheng. ✉e-mail: qli@fudan.edu.cn; shxwang@tsinghua.edu.cn

decades[13–15]. PMs emitted from industrial sectors have gradually decreased after decades of emission control policy implementation in major economies, including the USA, Europe, and China. However, the current regulations aimed at reducing the PM emission quantity cannot proportionally reduce the adverse health effects of industrial PM emissions.

The heterogeneity in source-specific PMs, including the chemical composition and toxicological properties, and the toxicity effects of PMs, such as the oxidative stress (OS) potency, have been verified as more health-relevant metrics than the PM total mass[16,17]. The components of source-specific PMs, which comprise complex mixtures of chemicals[3,18], vary widely among industrial sectors[19–21], owing to the notable discrepancies in raw materials, boiler facilities and/or production processes, as well as air pollution control technologies[22–24]. Different types of chemical components exhibit varying biological effects[25]. Transition metals and polycyclic aromatic hydrocarbons (PAHs) can lead to the generation of reactive oxygen species (ROS), thus yielding negative health outcomes[17,18,26], while many of the major PM components, including non-toxic inorganic sulfate and nitrate, might have interactive effects with the known toxic species[27]. Thus, the toxic potencies of PMs emitted from different industrial sources, could likely exhibit a notable discrepancy. However, toxic components such as trace elements and PAHs and their roles in determining the toxic potency of source-specific PMs have been scarcely identified and quantified[28–30]. Furthermore, the toxic potencies of PMs emitted from different industrial sectors have been scarcely investigated based on field study. The lack of information on source-specific chemicals and relevant chemical-specific toxicities limits the current understanding of the adverse impacts of exposure to industry-emitted PM, which also hinders the formulation of effective control strategies for human health protection.

In the present study, optimal health-relevant air pollution control strategies for industrial plants are proposed by integrating source-specific toxic potencies and pollution reducing costs. The three most polluting industrial sectors, i.e., the iron and steel industry (ISI), cement industry, and power industry, were selected in this work. $PM_{2.5}$ samples (refer to the filterable $PM_{2.5}$, fine particles that are directly emitted as a solid or liquid phase at the stack) were collected from these three sectors in field across 18 provinces in mainland China (Supplementary Fig. 1 and Supplementary Tables 1−3), which exhibit the highest production and consumption levels of fossil fuel-fired power, steel, and cement worldwide[24,31]. The obtained chemical composition and toxicological data (Supplementary Figs. 2−5 and Supplementary Table 4) were further integrated with the model-simulated PM mass concentration under the ultralow emission (ULE) scenarios, which was estimated by using the updated Air Benefit and Cost and Attainment Assessment System (ABaCAS) emission inventory combined with the Weather Research and Forecasting−Community Multiscale Air Quality (WRF-CMAQ) model[32–34]. The integrated results could provide insight into the adverse effects of population exposure to industry-emitted PMs and could facilitate the formulation of priority control strategies to achieve cost-effective reduction in toxic potency-adjusted PM exposure risks. The experiments and analysis approach are detailed in the Methods section.

## Results And Discussion

### Unequal toxic potencies of industry-emitted $PM_{2.5}$

Exposure to different industry-emitted $PM_{2.5}$ samples exhibits substantial discrepancies in toxic potencies, including intracellular OS and cytotoxicity (CT) potencies (Fig. 1 and Supplementary Fig. 2). $EC_{1.5}$ (the concentration effectively inducing a 1.5-fold production of intracellular ROS generation relative to that of the control) and $IC_{20}$ (the concentration effectively inducing a 20% inhibition relative to that of the control) values were determined to elucidate the endpoints of $PM_{2.5}$-induced OS and CT potencies, respectively. The toxic potency increased with decreasing $EC_{1.5}$ and $IC_{20}$ values. ISI-emitted $PM_{2.5}$

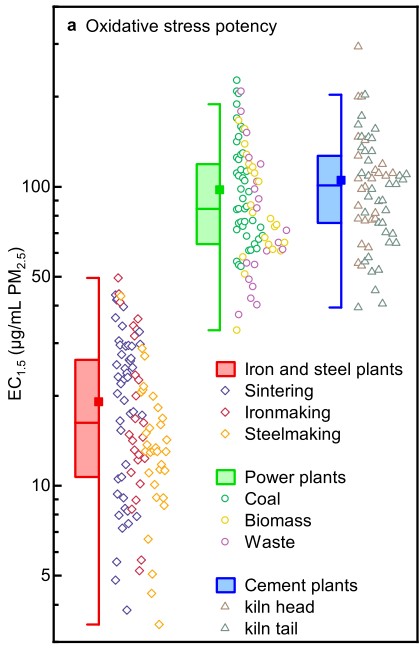
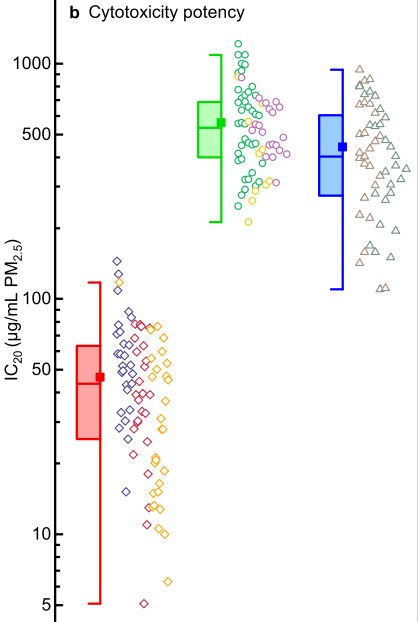
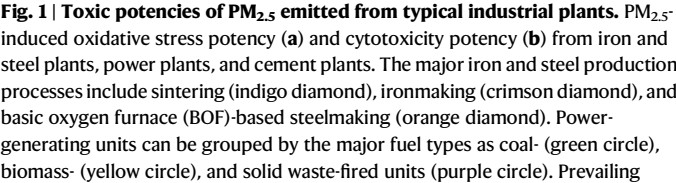

**Fig. 1 | Toxic potencies of $PM_{2.5}$ emitted from typical industrial plants.** $PM_{2.5}$-induced oxidative stress potency (**a**) and cytotoxicity potency (**b**) from iron and steel plants, power plants, and cement plants. The major iron and steel production processes include sintering (indigo diamond), ironmaking (crimson diamond), and basic oxygen furnace (BOF)-based steelmaking (orange diamond). Power-generating units can be grouped by the major fuel types as coal- (green circle), biomass- (yellow circle), and solid waste-fired units (purple circle). Prevailing processes of pollutant emissions in clinker and cement production are kiln head (brown triangle) and kiln tail (olive triganle). The toxic potencies of $PM_{2.5}$ increased with decreasing $EC_{1.5}$ and $IC_{20}$ values. Data plotted as box-and-whiskers (The line of the box is median line, the solid squares are mean values, the edges of the boxes are the quartile range (25th and 75th percentile), and the whisker is range within 1.5 times from the interquartile range).

exhibited the highest OS potency ($19.1 \pm 10.9$ µg/mL), which is nearly 5.0 and 5.4 times higher than that of power industry-emitted $PM_{2.5}$ ($94.9 \pm 41.9$ µg/mL) and cement industry-emitted $PM_{2.5}$ ($103.6 \pm 46.3$ µg/mL), respectively (Fig. 1a).[23] The $EC_{1.5}$ values for the $PM_{2.5}$ originating from the three major polluting processes of basic oxygen furnace (BOF)-based crude steel production, i.e., the sinter ore ($21.3 \pm 11.2$ µg/mL), pig iron ($20.7 \pm 12.1$ µg/mL), and crude steel ($14.7 \pm 7.8$ µg/mL) generation, are shown in Supplementary Fig. 2b, while the $EC_{1.5}$ values for other ISI processes (including pelletizing, electric arc furnaces (EAF)-based steelmaking, and steel rolling) are shown in Supplementary Fig. 3. The three processes contributed nearly 90% to the nationwide ISI $PM_{2.5}$ emissions in mainland China in 2019[31]. The averaged OS potency of power industry-emitted $PM_{2.5}$ resulted in very close $EC_{1.5}$ values when the boilers were fed with different solid fuels, including coal ($104 \pm 44$ µg/mL), solid waste ($89.7 \pm 48.1$ µg/mL), and biomass ($91.9 \pm 35.7$ µg/mL) (Supplementary Fig. 2c). Additionally, the $EC_{1.5}$ values for the $PM_{2.5}$ originating from the major production processes of cement plants were close, at $113.4 \pm 53.0$ µg/mL for the kiln head and $98.7 \pm 39.9$ µg/mL for the kiln tail (Supplementary Fig. 2d).

The distribution of the $IC_{20}$ values further indicated similar substantial discrepancies among the three industrial sectors. The estimated CT potency of ISI-emitted $PM_{2.5}$ ($46.4 \pm 28.2$ µg/mL) was nearly 12.1 times that of power generation-emitted $PM_{2.5}$ ($561 \pm 226$ µg/mL) and 9.5 times that of cement production-emitted $PM_{2.5}$ ($442 \pm 221$ µg/mL) (Fig. 1b). The CT potency of ISI-emitted $PM_{2.5}$ successively increased for the major production processes of sinter ore ($60.0 \pm 30.4$ µg/mL), pig iron ($43.4 \pm 22.7$ µg/mL), and crude steel ($36.1 \pm 26.4$ µg/mL) (Supplementary Fig. 2f). The $IC_{20}$ values for the other ISI processes are shown in Supplementary Fig. 3. The CT potency of power industry-emitted $PM_{2.5}$ slightly varied among the different fuel-fired units, including coal- ($604 \pm 254$ µg/mL), biomass- ($434 \pm 212$ µg/mL), and solid waste-fired units ($539 \pm 137$ µg/mL) (Supplementary Fig. 2g). In terms of clinker production, the CT potency of kiln tail emitted-$PM_{2.5}$ ($396 \pm 214$ µg/mL) was also close to that of kiln head-emitted $PM_{2.5}$ ($487 \pm 221$ µg/mL) (Supplementary Fig. 2h).

Estimation through laboratory and field studies of all the major processes under the same quality control conditions revealed that the $PM_{2.5}$ emitted from the ISI is much more toxic than those of the power and cement industries, especially the $PM_{2.5}$ emitted from blast furnace (BF)-based ironmaking and BOF-based steelmaking processes. A substantial inequality in the toxicological properties of the $PM_{2.5}$ samples obtained from the three industrial sectors is uncovered according to the estimated OS and CT potencies. Source-specific $PM_{2.5}$ comprises a complex mixture of inorganic and organic chemicals, some of which are potential contributors to $PM_{2.5}$ related toxicity[17,18]. Hence, it is necessary to uncover and quantify the key chemical components that determine $PM_{2.5}$ toxicological effects to better understand the causes of the abovementioned inequality in toxicological properties.

## Substantial discrepancies among key toxic chemicals in $PM_{2.5}$

The chemical composition, including carbonaceous species (i.e., organic matters and elemental carbon), water soluble inorganic ions, and elements, widely varied among the $PM_{2.5}$ originating from the three industrial sources (Supplementary Fig. 4 and Supplementary Table 4). The role of EC was not taken into consideration due to its minor mass concentration in per unit $PM_{2.5}$ mass and the relative lower toxic potency compared to PAHs and metals. Here, the targeted chemicals are focused on metals and PAHs, which are commonly regarded as key toxic components associated with the biological effects of $PM_{2.5}$[17,18]. Ten toxic metals (i.e., V, Cr, Mn, Fe, Ni, Cu, Zn, As, Cd, and Pb) were observed at substantially higher concentrations per unit mass of ISI-emitted $PM_{2.5}$ ($253 \pm 106$ mg/g), exceeding those per unit mass of power industry- ($14.9 \pm 4.5$ mg/g) and the cement industry- ($23.5 \pm 7.8$ mg/g) emitted $PM_{2.5}$ by approximately 17 and 11 times,

respectively (Fig. 2a). Fe dominated the mass concentrations, with relative concentrations ranging from 62.6–98.3%, 43.0–94.8%, and 80.4–94.1% in the selected metals within $PM_{2.5}$ emitted from the ISI, power, and cement industries, respectively (Fig. 2a). The observation results regarding the variation in toxic metals are consistent with those obtained in our previous study in regard to flue gases emitted from industrial plants[35]. The concentrations of metals per unit mass of $PM_{2.5}$ emissions grouped by the prevailing processes in iron and steel production, the fuel type in power-generating units, and the prevailing processes in clinker and cement production are detailed in Supplementary Note 7 and Supplementary Fig. 5a. The targeted toxic metals of concern predominantly exist in the particulate phase, not the vapor phase, in typical industrial flue gases. This is mainly due to their substantially higher gasification temperatures[36–38]. Consequently, the condensation of these maters after being discharged from the stack can be disregarded or considered negligible.

Other key toxic chemical in $PM_{2.5}$, i.e., 16 US Environmental Protection Agency (EPA) priority PAHs (i.e., Nap, Acy, Ace, Flu, Phe, Ant, Flt, Pyr, BaA, Chry, BbF, BkF, BaP, InP, DahA, and Bghip), were also revealed to substantially differ among the three industrial sectors studied (Fig. 2b). The content of these 16 PAHs ($686 \pm 775$ µg/g) in mass-normalized ISI-emitted $PM_{2.5}$ was one order of magnitude higher than that in mass-normalized power sector-emitted $PM_{2.5}$ ($27.7 \pm 39.2$ µg/g) and cement sector-emitted $PM_{2.5}$ ($14.6 \pm 14.8$ µg/g). Compared to the PAHs emitted from the power and cement industries, higher carcinogenic species (with toxic equivalent factor (TEF) equal to or greater than 0.1) were more abundant in ISI-emitted PAHs, accounting for 31.2–37.8% of the total mass of the 16 PAHs. In contrast, two- and three-ring PAHs (species with a lower toxicity) were the dominant species among the PAHs emitted from power and cement industries, owing to the higher combustion efficiency of their practical boilers than that of the boilers at ISI plants. The corresponding toxic equivalent benzo[a]pyrene values ($BaP_{eq}$) of the 16 PAHs can directly reflect their potential toxic discrepancies (Supplementary Fig. 6). The $BaP_{eq}$ value per unit mass of ISI-emitted $PM_{2.5}$ ($44.5 \pm 78.5$ µg/g) was two orders of magnitude higher than that per unit mass of power industry-emitted $PM_{2.5}$ ($0.21 \pm 0.23$ µg/g) and cement industry-emitted $PM_{2.5}$ ($0.51 \pm 0.68$ µg/g). Thus, the PAHs originating from the iron and steel production posed a much higher carcinogenic risk than those originating from the other two industrial sectors. The concentrations of the 16 PAHs per unit of $PM_{2.5}$ emitted from individual ironmaking and steelmaking processes, power generation units driven by different fuel types, and individual clinker- and cement-making processes are detailed in the Supplementary Note 7 and Supplementary Fig. 5b.

The relative contributions of the targeted metals and PAHs to $PM_{2.5}$-induced OS potency, quantified via the concentration addition reference model[39], are shown in Supplementary Fig. 7. The $PM_{2.5}$-induced OS potency contributed from metals were estimated basing on the tested chemicals performed in the highest valance due to the majority of them are usually in the form of the highest valence in $PM_{2.5}$[40,41]. The targeted metals contributed $84.6 \pm 22.6\%$, $83.8 \pm 9.8\%$, and $77.5 \pm 8.4\%$ to OS potency induced by the primary $PM_{2.5}$ emitted from the ISI, power, and cement industries, respectively. While the identified PAHs accounted for minor fractions (2.7–3.6%) of the total $PM_{2.5}$-induced OS potency among the three major sectors (Supplementary Fig. 7a). The results indicated that the $PM_{2.5}$-induced OS potency could mainly be attributed to $PM_{2.5}$-bound toxic metals. The higher concentration of toxic metals per unit mass of ISI-emitted $PM_{2.5}$ led to a much higher ROS-inducing effect than that of the $PM_{2.5}$ emitted from the power and cement industries. Thus, the unequal toxic potencies of industry-emitted $PM_{2.5}$ could be attributed to the substantial discrepancies among toxic chemicals in $PM_{2.5}$ emitted from the different industrial sectors (Supplementary Fig. 7b and c). The toxicity results observed from industrial PM are consistent with epidemiological studies, which indicate that transition metals are likely to

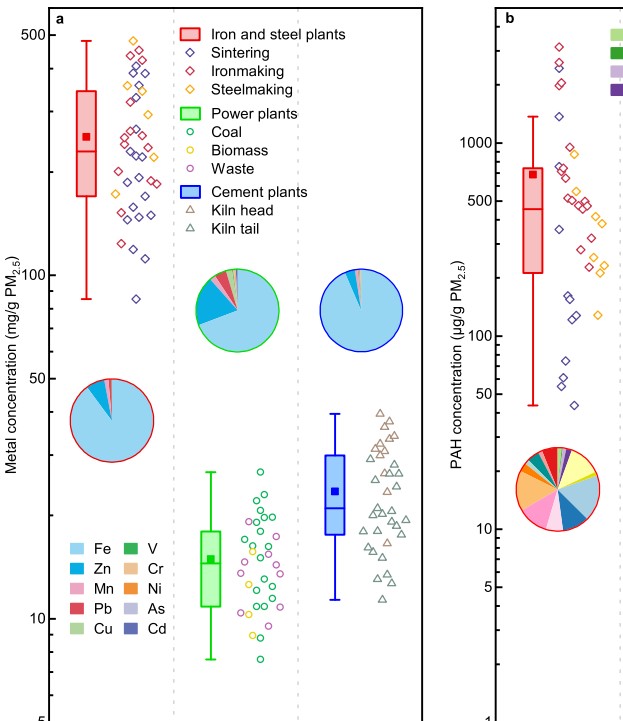

**Fig. 2 | Mass-normalized concentrations of the selected metals and PAHs contained in PM$_{2.5}$ originating from large-scale industrial plants.** Concentrations of (**a**) 10 selected toxic metals (i.e., V, Cr, Mn, Fe, Ni, Cu, Zn, As, Cd, and Pb) and (**b**) 16 US EPA priority PAHs (i.e., Nap, Acy, Ace, Flu, Phe, Ant, Flt, Pyr, BaA, Chry, BbF, BkF, BaP, InP, DahA, and Bghip) per unit mass of PM$_{2.5}$ emitted from typical plants in the iron and steel, power, and cement industries. The tested plants in the three industrial sectors are conventionally grouped by the rude steel production processes, fuel types, and clinker production processes, respectively. Data plotted as box-and-whiskers (The line of the box is median line, the solid squares are mean values, the edges of the boxes are the quartile range (25th and 75th percentile), and the whisker is range within 1.5 times from interquartile range). The mass fractions of individual metals and PAHs are shown as pie charts.

be responsible for the adverse effects of PMs, including induce oxidative stress, and impairment in lung function[42,43]. Furthermore, multiple epidemiological evidence also shows the association between industrial PM emissions and adverse health outcomes[44–46]. On the global scale, industrial sector emissions roughly account for 11.7% of the PM$_{2.5}$ disease burden[44]. Then the quantified toxic potencies of industries-emitted PM$_{2.5}$ could be integrated with emission inventories to develop toxic potency-adjusted emission policies toward healthy air.

**The iron and steel industry dominates PM$_{2.5}$ toxic potency-adjusted emissions**

The quantified toxic potencies were integrated into PM$_{2.5}$ emission inventories as PM$_{2.5}$-induced toxic potency-adjusted emissions of the three selected industrial sectors in mainland China from 2005–2019, as shown in Fig. 3. As the largest contributor to PM$_{2.5}$-induced toxic potency-adjusted emissions, the ISI is grouped by its major processes, including sintering, ironmaking, BOF-based steelmaking, and other processes. There existed substantial disproportionalities between the PM$_{2.5}$ emissions and PM$_{2.5}$-induced toxic potency-adjusted emissions of the three industrial sectors. ISI-emitted PM$_{2.5}$ accounted for only 29.0 ± 7.6% of nationwide PM$_{2.5}$ emissions among the three sectors in 2005, but accounted for approximately 70.7 ± 32.9% of PM$_{2.5}$-induced OS potency-adjusted emissions (Fig. 3a). While power and cement industries-emitted PM$_{2.5}$ accounted for 71.0 ± 16.4 % of the total mass emissions, and only 29.3 ± 10.4% of the total PM$_{2.5}$-induced OS potency-adjusted emissions. Owing to the drastic reduction in PM$_{2.5}$ emitted from power industry under the enforcement of the toughest ULE standards since 2014[22], the relative fractional contribution of the ISI increased to 43.8 ± 12.0% of mass emissions and further contributed 81.8 ± 37.3% to the overall PM$_{2.5}$-induced OS potency adjusted-emissions in 2019 (Fig. 3b).

The trends in total PM$_{2.5}$-induced OS potency-adjusted emissions of the three industrial sectors between 2005 and 2019, as illustrated in Fig. 3c, substantially differed from the corresponding trends in PM$_{2.5}$ emissions (Supplementary Fig. 8). The total PM$_{2.5}$-induced OS potency-adjusted emissions fluctuated and eventually increased by approximately 12.2% between 2005 and 2012, although the total amount of nationwide PM$_{2.5}$ emitted from the three sectors decreased by 20.0% during the same period. The increase in OS potency-adjusted emissions was mainly attributable to the ISI emissions increase, particularly the emissions originating from BOF-based steelmaking and sintering processes. Since the per unit mass of ISI-emitted PM$_{2.5}$ exhibit much higher OS toxic potencies, the slight increase (~10.4%) in ISI-emitted PM$_{2.5}$ resulted in an increase of ~12.2% in the total PM$_{2.5}$-induced OS potency-adjusted emissions. The reduced OS potency attributed to the reduction in PM$_{2.5}$ emissions in the power (629 kt) and cement (1462 kt) sectors was offset by the slight increase (716 kt) in the ISI-emitted PM$_{2.5}$. The PM$_{2.5}$-induced OS potency-adjusted emissions of the three sectors decreased by ~65.1% between 2012 and 2019. The decreases in the toxic potencies were mainly driven by the corresponding reductions in the ISI sector, with a contribution of ~86.8% to OS potency-adjusted emissions. While the reductions in the power generation and cement production sectors jointly contributed only 13.2% to the overall reduction in PM$_{2.5}$-induced OS potency-adjusted emissions, although these two sectors jointly accounted for ~47.2% of the reduction in total PM$_{2.5}$ mass emissions.

The temporal changes in PM$_{2.5}$-induced CT potency-adjusted emissions in the three industrial sectors exhibited a similar tendency to those in OS potency-adjusted emissions, as shown in Supplementary Fig. 9. Substantial reductions in the PM$_{2.5}$ mass-based emissions of the power industry, which could be attributed to the implementation of the toughest-ever ULE standards based on the PM mass concentration, do not necessarily suggest an equivalent alleviation of PM$_{2.5}$ health

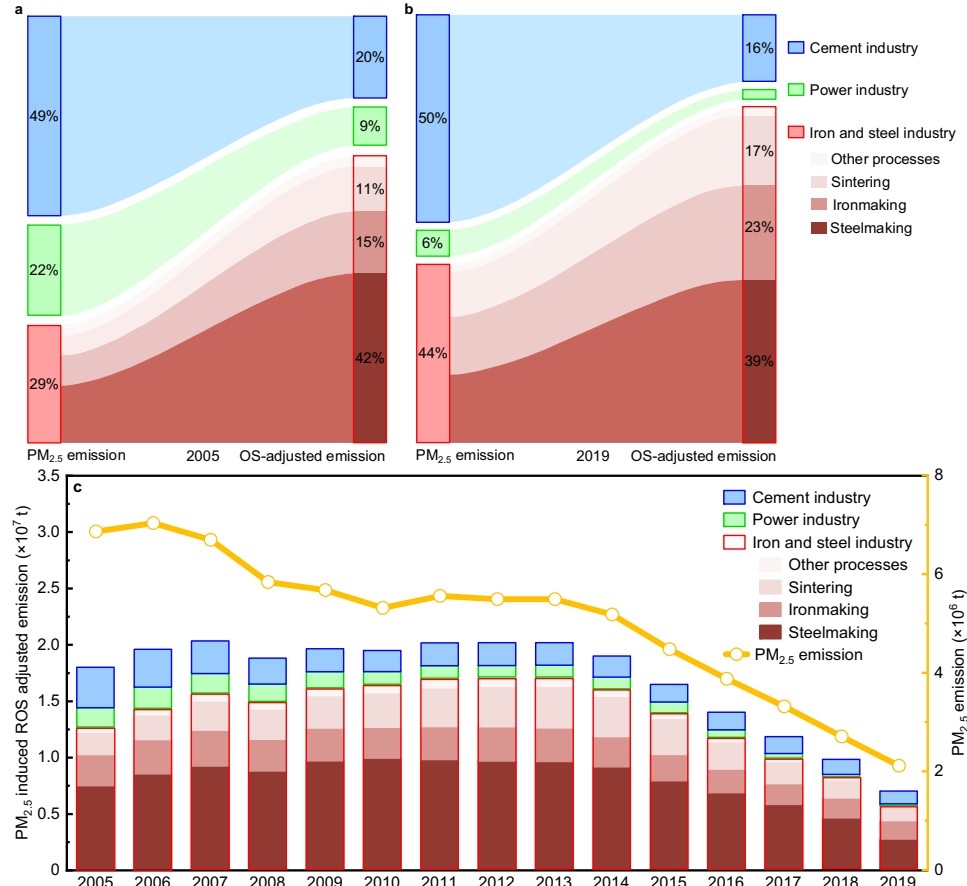

**Fig. 3 | Changes in the PM$_{2.5}$ emissions and PM$_{2.5}$-induced oxidative stress potency-adjusted emissions of the three industrial sectors in China.** Relative distribution of the PM$_{2.5}$ emissions and the induced oxidative stress (OS) potency-adjusted emissions of the three industrial sectors (i.e., iron and steel production, power generation, and cement production) in 2005 (**a**) and 2019 (**b**), and temporal evolution the of PM$_{2.5}$ absolute mass (orange line) and PM$_{2.5}$-induced OS potency-adjusted mass emissions from the three industrial sources between 2005 and 2019 (**c**).

effects, as evidenced, for example, by the toxic potency (including OS and CT potencies)-adjusted emissions in the present study. In contrast, the slight increases or decreases in ISI-emitted PM$_{2.5}$ substantially dominated the variations in PM$_{2.5}$ toxic potency-adjusted emissions among the three sectors. The disproportionate variation in nationwide PM$_{2.5}$ toxic potency-adjusted emissions was mainly due to the unequal source-specific toxic potencies at equal PM$_{2.5}$ mass concentrations. Furthermore, regional discrepancies in the industrial structure among the three sectors could lead to spatial variations in PM$_{2.5}$ emissions and toxic potency-adjusted PM$_{2.5}$ exposure risks nationwide.

**Priority control in the iron and steel industry can result in a cost-effective reduction in the population-weighted PM$_{2.5}$ exposure risk**

Two reduction routes were developed to quantify the reduction effects on mass-based industrial PM$_{2.5}$ emissions and toxic potency-adjusted industrial PM$_{2.5}$ emissions (including the estimation of OS and CT potencies) if all facilities in power, cement, and iron and steel industries meet the ULE levels, respectively (Fig. 4a). The implementation of ULE standards in the power industry (1000 kt) could yield a nearly 1.5- and 1.6-fold higher reduction in nationwide mass-based PM$_{2.5}$ emissions than that obtained via the implementation of ULE standards in the ISI (621 kt) and cement industry (653 kt), respectively. However, the implementation of ULE standards in the ISI could lead to the largest reduction in PM$_{2.5}$-induced toxic potency-adjusted emissions (including the estimation of OS and CT potencies) among the three industrial sectors owing to the unequal OS and CT potency at an equal PM$_{2.5}$ mass, respectively. Spatial distributions of

the ambient PM$_{2.5}$ concentration contributed by primary PM$_{2.5}$ originating from the three sectors before and after ULE standards implementation in each sector are shown in Supplementary Fig. 10. The nationwide primary PM$_{2.5}$ concentrations originating from the three industries if all existing units in the power, cement, and iron and steel industries, respectively, satisfied ULE standards in 2019 are shown in Supplementary Fig. 11. The nationwide annual mean population-weighted PM$_{2.5}$ exposure could decrease by 37.6%, and 26.0%, and 36.4% if all facilities in power, cement, and ISI, respectively, met ULE standards in 2019 (Supplementary Table 5). The toxic potency-adjusted exposure risk (TPAE) metric was further developed to estimate a health-relevant risk index for population exposure to industrial emitted PM$_{2.5}$, including estimation of PM$_{2.5}$-induced OS potency- and CT potency-adjusted exposure risks (TPAE$_{OS}$ and TPAE$_{CT}$). The cited toxic equivalent values of PM$_{2.5}$-induced OS and CT potencies are provided in Supplementary Table 6.

Figure 4b–d show the spatial distributions of the corresponding reduction in the total TPAE$_{OS}$ value attributed to meeting ULE standards in the power, cement, and ISI in 2019, respectively. The decrease in the power and cement industries only accounted for a minor faction of the total reduction in the nationwide TPAE$_{OS}$ value if all their units achieved ULE standards in 2019 (Fig. 4b and c). The highest reductions in power industry- and cement industry-related TPAE$_{OS}$ could be occurred in north and central China, respectively, which were consistent with the density of power and cement plants. While the substantial reduction in the nationwide TPAE$_{OS}$ value could be obtained if all ISI facilities satisfied ULE standards in 2019 (Fig. 4d). Much higher reductions in ISI-associated TPAE$_{OS}$ could be observed on the northern

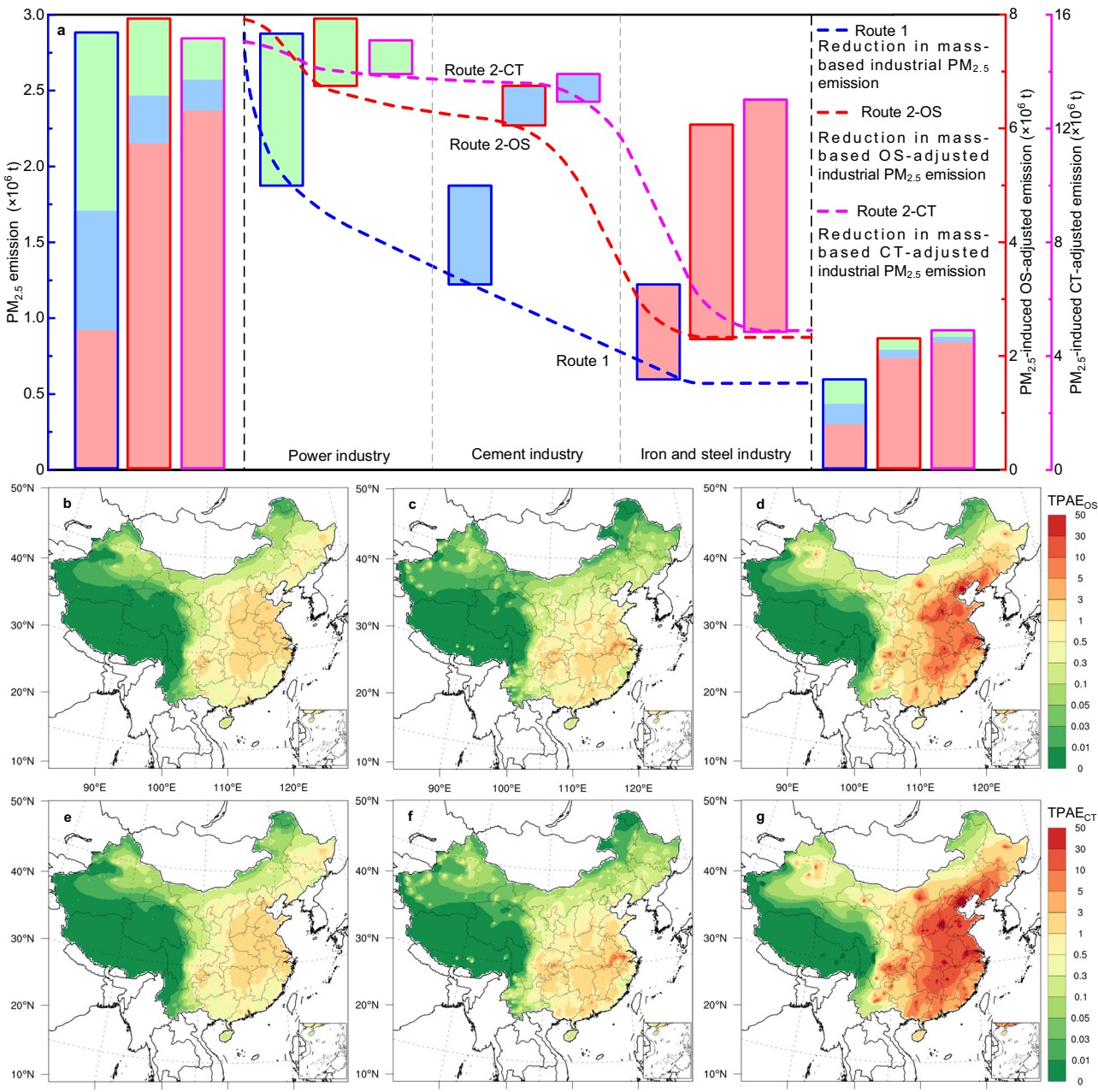

**Fig. 4 | Reduction in the industrial PM$_{2.5}$ emissions and PM$_{2.5}$-induced toxic potency adjusted exposure under ultralow emission standards in China.** Two reduction routes (including route 1 toward PM$_{2.5}$ mass-based emission reduction and route 2 toward oxidative stress (OS) potency- and cytotoxicity (CT) potency-adjusted emissions reduction) under the implementation of ultralow emission standards in each industrial sector (i.e., iron and steel production, power generation, and cement production) (**a**), as well as spatial distributions of the corresponding reductions in the OS potency-adjusted exposure (TPAE$_{OS}$) and CT potency-adjusted exposure (TPAE$_{CT}$) attributed to all facilities meeting the ultra-low emission standards in the power industry (**b, e**), cement industry (**c, f**), and iron and steel industry (**d, g**) in 2019, respectively. The maps (**b–g**) were provided by NCAR Command Language (NCL)[63].

and eastern China, especially in densely populated areas such as Hebei and Jiangsu provinces. Since these populated areas concentrated with a large portion of ironmaking and steelmaking plants, and majority of these plants could attain ULE levels due to the priority implementation of ULE standards and a range of advanced technologies[23]. TPAE$_{OS}$ values attributed to each industry source with scenarios before and after ULE standards introduction are shown in Supplementary Fig. 12. The implementation of ULE standards in the power industry yielded the largest reduction in nationwide PM$_{2.5}$ emissions (Fig. 4a) and PM$_{2.5}$ concentration (Supplementary Figs. 10 and 11) among the three industrial sources. However, the implementation of ULE standards in

the ISI sector could dominate the reduction in the nationwide TPAE$_{OS}$ value among the three sectors if the unequal OS potencies of source-specific PM$_{2.5}$ were included (Supplementary Table 4).

The nationwide total population-weighted TPAE$_{OS}$ in the three sectors was dominated by the ISI (83%), owing to the higher OS potency per unit mass of ISI-emitted PM$_{2.5}$. The implementation of ULE standards in the ISI sector could generate 5.4 ± 4.0 and 8.2 ± 6.6 times higher reduction in the nationwide population-weighted TPAE$_{OS}$ than that obtained via the implementation of ULE standards in the power and cement industry sectors in 2019, respectively (Supplementary Table 7). The reduction in PM$_{2.5}$-induced CT potency-adjusted

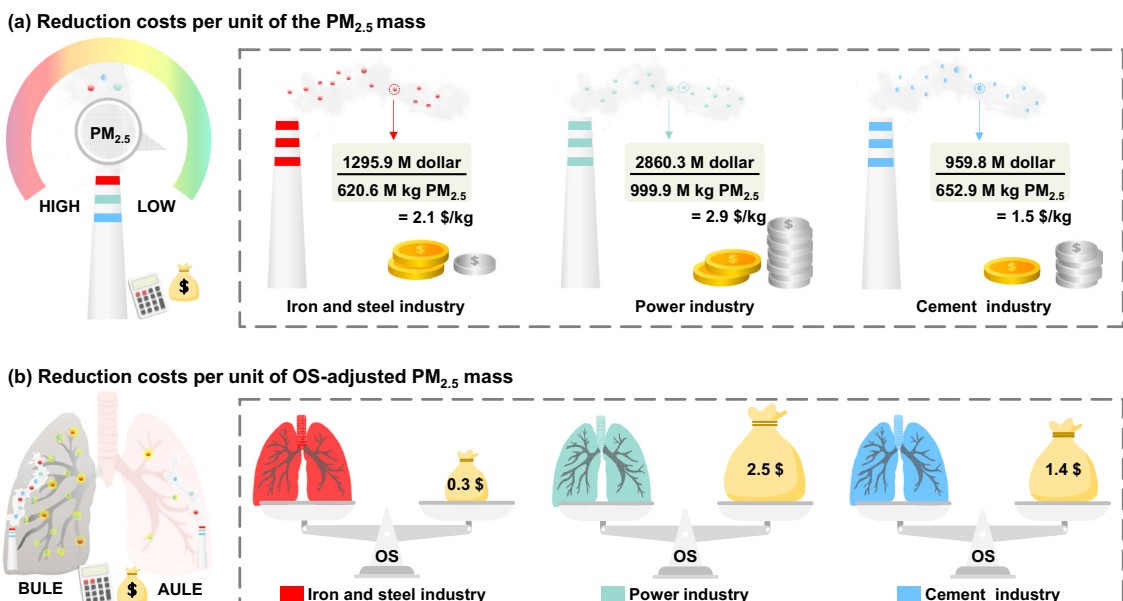

**Fig. 5 | Costs for the reductions in the PM$_{2.5}$ mass and PM$_{2.5}$-induced oxidative stress (OS) potency-adjusted mass from the industrial sectors.** Reduction costs per unit of the PM$_{2.5}$ mass (kg) (**a**) and per unit of the OS potency-adjusted PM$_{2.5}$ mass (**b**) if all facilities in the iron and steel industry (red), power industry (green), and cement industry (blue) meet the ULE standards, respectively. The unit cost, namely marginal reduction cost, indicates the ratio of additional increased investment and the PM$_{2.5}$ abatement potential under the application of the ULE standards to each industry source (as detailed in Methods section and Supplementary Note 10).

emissions and the reduction effect on the nationwide TPAE$_{CT}$ as well as population-weighted TPAE$_{CT}$ exhibited similar tendencies to that of TPAE$_{OS}$ in mainland China (Fig.4a and e–g, Supplementary Fig. 13 and Supplementary Table 7). The results indicate that implementation of ULE standards in the ISI sector can achieve the largest reduction effect in both nationwide TPAE$_{OS}$ and TPAE$_{CT}$ in the three industrial sectors. Furthermore, the marginal reduction costs for PMs emitted from industries can be integrated into the analysis when installation and operating costs of dedusting systems to meet ULE standards are considered.

Figure 5 shows the marginal reduction costs for per unit of industry-emitted PM$_{2.5}$ mass and per unit of toxic potency-adjusted industry emitted-PM$_{2.5}$ mass under the scenario with all facilities in each industrial sector meeting ULE standards across the nation in 2019. The marginal abatement cost per ton of the PM$_{2.5}$ mass of the ISI sector was nearly 73% and 142% that per ton of the PM$_{2.5}$ emitted from the power and cement industries (Fig. 5a), respectively, while the abatement cost for per unit of OS potency-adjusted mass of ISI-emitted PM$_{2.5}$ was only 14% and 25% that for per unit of OS potency-adjusted mass of power- and cement-emitted PM$_{2.5}$ (Fig. 5b), respectively. The marginal cost for abating per unit mass of the CT potency-adjusted ISI-emitted PM$_{2.5}$ is only 12% and 15% that for per unit mass of CT potency-adjusted power- and cement-emitted PM$_{2.5}$ (Supplementary Fig. 14), respectively. The cost estimates indicated that reductions in the emission sources of higher toxic PM$_{2.5}$ could greatly benefit the decrease in PM$_{2.5}$-induced OS potency- and CT potency-adjusted mass emissions instead of the reduction in the largest PM$_{2.5}$ emission source. Thus, we suggest that priority control in the ISI could achieve a much more cost-effective reduction in the adverse impacts associated with exposure to PM$_{2.5}$ originating from the three industrial sources.

## Discussion and policy implication

The unequal toxic potencies of primary PM$_{2.5}$ emitted from the different industrial sources, including the ISI, power, and cement industries, were elucidated via real-world measurements and laboratory analysis under the same quality control conditions (including field and procedure blank correction). Oxidative stress is established as one of the important mediators leading to adverse health outcomes of air pollution. Therefore, oxidative potential is a good and robust starting point to compare the mass-based and health-oriented frameworks in both technical and economic considerations[16]. Furthermore, increasing number of epidemiological studies have found that OP was more strongly associated with cardiorespiratory disease than PM$_{2.5}$ mass[47,48], indicating that OP may be a more health relevant health metric than PM$_{2.5}$ mass. However, this health-relevant metric is still absent in current control policies based on pollutant mass concentrations worldwide[3,16]. As the conventional top polluting industry ranked by the total PM mass emissions, the toughest control policies were primarily enforced at power plants. However, the reduction in the nationwide TPAE value attributed to the implementation of ULE control policies at power plants accounted for only 6.6–13.1% of total TPAE reduction among the three industrial sectors across mainland China, owing to the low toxic potency per unit mass of power industry-emitted PM$_{2.5}$. In contrast, the decrease in the ISI after the implementation of ULE standards at all facilities dominated the nationwide reduction in the total TPAE value (76.2–86.7%) among the three sectors in 2019 across mainland China. Furthermore, the marginal abatement cost for the mitigation of per unit mass of toxic potency-adjusted ISI-emitted PM$_{2.5}$ is only 12–25% that for the mitigation of per unit mass of toxic potency-adjusted power industry-emitted PM$_{2.5}$ under the scenario whereby all ISI facilities in mainland China reach satisfied standards in 2019 (Fig. 5). Therefore, the ISI should be prioritized among the three sectors, given its cost-effective role in reducing adverse impacts associated with exposure to industry emitted-PM$_{2.5}$. We highlight that it is urgent to shift from the mass-based control strategy to the health-oriented control policies via integrating unequal toxicities of industrial particulate emissions.

The implementation of stricter control measures targeting ISI-emitted PM$_{2.5}$ could be a big challenge in practice because iron and steel production involves complex manufacturing processes and

multiple emission sources[23], including the major processes of sintering, ironmaking, and BOF-based steelmaking. Due to their highest PM$_{2.5}$ emissions and highest PM$_{2.5}$-induced toxic potency values, these major processes represent important targets for further adverse impacts reduction. Additionally, compared to the effects of conventional BOF-based steelmaking processes, EAF-based steelmaking processes could alleviate PM$_{2.5}$-induced toxic potencies, as indicated by our limited EAF-emitted PM$_{2.5}$ samples (Supplementary Fig. 3). The obtained toxicity results expand the previous recognition of EAFs as an advanced technology option for the mitigation of ISI-emitted air pollutants[31]. The shift from BOFs to EAFs is recommended over the next decades to achieve carbon neutrality and reduce ISI-emitted air pollutants, especially in high income regions. The uncertainties in the toxicity of EAF-emitted PM$_{2.5}$ may hinder the evaluation of the effectiveness of current EAF technology in addressing the health effects of PM$_{2.5}$ pollution. Hence, future efforts are warranted to quantify and clarify the toxic potency of EAF-emitted PM$_{2.5}$. The obtained understanding may offer important insights to policymakers formulating targeted control policies to achieve air quality, climate, and health cobenefits in the ISI.

Several limitations and uncertainties in field investigations and estimations exist in this study. This study focused on the filterable PM (refers to particles that are directly emitted as a solid or liquid at stack conditions), while condensable PM (the material in vapor phase at the stack but condenses to form solid or liquid PM immediately after discharge from the stack into the ambient air) originating from industrial sectors, as well as source-specific secondary and fugitive aerosols emission from manufacturing iron/steel and cement processes, were not considered in this study, which could result in underestimation of the adverse impacts associated with exposure to specific source-emitted aerosols. The observed toxic potencies based on in vitro cell tests are not strictly consistent with the actual PM$_{2.5}$ exposure environment in the human body, and thus, toxic estimation methods limit the understanding of real-world human exposure scenarios. In future, it is necessary to investigate the potential toxicities of industrial condensable PMs and explore the quantitative connection between their toxic potencies and the corresponding chemical composition. Furthermore, quantifying the role of condensable PMs in adjusting the toxicity of filterable PMs is crucial for a comprehensively understanding of the adverse impacts originating from industry sources. Additionally, the industry source-emitted secondary and fugitive aerosols and their induced toxic potencies are encouraged to be quantified to comprehensively estimate the adverse impacts associated with exposure to industry-emitted PM$_{2.5}$. Wet plumes, formed when saturated wet flue gas is discharged from stack and enters to the atmosphere, are frequently observed in industrial plants[49]. However, the toxicity of wet plumes remains unclear, emphasizing the need for systematic studies to address these issues in the future. Toxicological studies (in vitro and in vivo tests) are urgently needed to quantify the interactive effects of individual chemical, such as sulfate and transition metals, and to clarify and validate the associations by integrating with epidemiological studies. In addition, more realistic toxicological exposure studies, including human organoid-based in vitro model, are required to elucidate the multiple health-relevant end points associated with exposure to industry-emitted PM$_{2.5}$. In vivo and epidemiological studies are strongly encouraged to clarify and validate the relationship between exposure to industry-emitted PM$_{2.5}$ and adverse health outcomes. On the other hand, corresponding ambient air PM samples may be collected in future studies for better risk assessment for large populations. The mixture effects of ambient PM$_{2.5}$, particularly in industrial areas, should be determined to better characterize and to dissect the quantitative contribution of toxic components and finally elucidate contributions of hazardous sources to mixture effects. Multilevel evidence could shed light on reducing the uncertainty in the estimation of the health effects of industry PM$_{2.5}$ pollution and the

formulation of tailored industry source control policies for health protection.

Despite the above limitations, our analysis could offer a perspective to policymakers, thereby emphasizing that a focus shift from emission mass-oriented policies to health-oriented control policies could help facilitate the most cost-effective reduction in the health effects of industry PM$_{2.5}$ pollution. The results may provide guidance for low and medium income regions suffering from similarly severe air pollution driven by multiple industrial sources that are seeking economic approaches for the mitigating of the health effects of industry-related emissions, such as India and Brazil[50]. Since the industry structure, manufacturing processes, and energy consumption exhibit large discrepancies among different regions worldwide, future work to quantify and rank the toxicity of individual industry source-emitted PM$_{2.5}$ (including condensable PM$_{2.5}$) are strongly recommended for further quantitative cost-benefit analysis and finally to formulate cost-effective control strategies to effectively mitigate source-specific PM$_{2.5}$ effects on public health.

## Methods
### Field measurements
Field studies of eighty-two typical units of energy- and emission-intensive industrial sectors, including the iron and steel industry (ISI), power industry, and cement industry, were conducted in 18 provinces in China from 2017–2022 (Supplementary Fig. 1). These selected units covered the major iron/steel and cement production processes, as well as the major types of fuel-fired power generating units. Information on these typical units is detailed in Supplementary Notes 1–4 and Supplementary Tables 1–3. The air pollution control measures enforced in the various industry sources, including the ISI, power, and cement plants, are outlined in Supplementary Note 1 and Supplementary Tables 8–11. An isokinetic sampling system (C-5000, ESC, USA) was deployed to collect PM$_{2.5}$ emission samples (namely, the filterable PM$_{2.5}$) originating from stationary sources following US EPA Method 17 and 201 A (Supplementary Note 5, Supplementary Fig. 15 and Supplementary Table 18)[21,51]. To prevent water vapor and fuel gas condensation, the deployed isokinetic sampling equipment was heated to and maintained at a temperature of 120 °C throughout the whole sampling process. Both quartz fiber and Teflon filters were employed to collect PM$_{2.5}$ samples for weighing and further analysis. The PM$_{2.5}$ remaining on the filter holders was removed by rinsing twice with acetone and collected in clean glassware for further weighing. The sampling time for PM$_{2.5}$ emitted from the three types of plants ranged from 30 to 120 min. The selected units were operated at over 75% of their capacity during each test to confirm a stable sampling condition. Three successful measurements of the selected units were performed at each sampling site. After sampling, each PM$_{2.5}$ sample was placed into a petri dish and sealed with a zipper bag and stored at −20 °C immediately before undergoing analysis.

### Chemical and toxic potency analyses
The chemical composition of PM$_{2.5}$ emitted from the three industrial sources, including organic carbon, elemental carbon, water soluble inorganic ions, PM$_{2.5}$-bound 16 US EPA PAHs, and elements, was quantified with the use of standard methods[21,49,52], as detailed in Supplementary Note 6. Gas chromatography coupled with mass spectrometry (Thermo Scientific ISQ 7000 GC – MS, USA) was employed to determine 16 EPA priority PAHs collected on quartz fiber filters. An inductively coupled plasma mass spectrometer (Thermo Scientific 7500a, USA) was used to examine elements collected on Teflon filters. BaP$_{eq}$ values were examined to estimate the carcinogenic risk of the 16 PAHs, which were estimated based on TEFs[53].

Human epithelial A549 cell lines were adopted to investigate PM$_{2.5}$-related toxic potencies, including PM$_{2.5}$-induced oxidative stress (OS) potency and cytotoxicity (CT) potency[52,54–56]. Descriptions of the

analysis process are provided in Supplementary Note 6. The OS potency induced by the $PM_{2.5}$ samples was examined with a 2′,7′-dichlorofluorescein diacetate protocol. The CT potency measurement was achieved via a 3-(4,5-dimethylthiazol-2-yl)-2,5-diphenyltetrazolium bromide assay. A microplate reader (Varioskan LUX, Thermo Scientific) was employed to investigate the fluorescence intensity and optical density, which were detected at wavelengths of 488/525 and 570 nm, respectively. The endpoints of $PM_{2.5}$-induced OS potency and CT potency were reported as $EC_{1.5}$ and $IC_{20}$ values, respectively.

## Development of the emission inventory and emission control scenarios

The ABaCAS emission inventory of air pollutants in China from 2005 to 2019 was developed on the basis of an emission-factor method, and the emissions from power plants and key industrial processes, such as the ISI and cement industry, were quantified via a unit-based approach[34,57,58]. Regarding the ISI, all specific industrial processes, including sintering, pelletizing, BF-based ironmaking, and BOF- and EAF-based steelmaking, were considered. Regarding the cement industry, cement grinding mills and cement clinkers, the latter of which include shaft kilns, rotary kilns, and new dry clinkers of different capacities, were considered. The activity data (energy consumption, product yield, etc.) from the energy statistical yearbooks and local or national surveys, such as the Second National Pollution Source Census, were collected[59]. For 2019, the detailed location, fuel type, energy consumption, product yields, and end-of-pipe control information of 5508 power plants, 742 BFs, 658 BOFs, 124 EAFs, and 1939 cement plants were collected for emission calculation and WRF-CMAQ simulation purposes[57].

Three control scenarios (no ultralow emission standard for electricity generation (ELE_noULE) scenario, ULE standard for cement industry (CI_ULE) scenario, and ULE standard for iron and steel industry (ISI_ULE) scenario) were developed to evaluate the potential emission reduction due to the implementation of ULE standards or technology improvement based on the base emission inventory of 2019, detailed as follows: (1) ELE_noULE scenario. Under this scenario, we used the energy consumption and technology structure of power plants in 2019 and the end-of-pipe control measures and shares of PM of power plants in 2014. (2) CI_ULE scenario. Under this scenario, we considered the product yield and technology structure of the cement industry in 2019 and end-of-pipe control technology of the ULE standard for PM control in the cement industry. (3) ISI_ULE scenario. Under this scenario, we employed the product yield and technology structure of the ISI in 2019 and end-of-pipe control technology of the ULE standard for PM control in the ISI.

## Estimation of the toxic potency-adjusted exposure

Ambient $PM_{2.5}$ concentrations were simulated with the WRF-CMAQ model. The simulation period covers the entirety of 2019. The WRF-CMAQ model with the updated ABaCAS emission inventory was adopted for baseline scenario simulation. The model simulation performance was evaluated via a comparison between the simulated parameter values and ground observations (Supplementary Note 8 and Supplementary Table 12). We operated the model with the above three emission control scenarios (including ELE_noULE, CI_ULE, and ISI_ULE) to estimate the reduction effect on toxic potency-adjusted exposure risk (TPAE) of the implementation of ULE standards and technology improvement. The differences between the baseline and ELE_noULE scenarios constitute the influence of the ULE standard for PM control implemented at power plants since 2015 in China. The differences between the baseline and CI_ULE and ISI_ULE scenarios represent the potential benefits of the implementation of ULE standards targeting the PM emissions from the cement industry and ISI, respectively.

Three additional zero-emission scenarios for the energy and industrial sectors were designed based on the emission inventory of 2019, including no electricity (NoELE) scenario, no cement industry (NoCI) scenario, and no iron and steel industry (NoISI) scenario, to examine the the contributions of power, cement, and ISI sectors to the ambient $PM_{2.5}$ concentration in the base year (Supplementary Table 13). Emission inventories excluding primary $PM_{2.5}$ emissions from power, cement, and ISI sectors, as obtained with the baseline scenarios, were used to simulate the NoELE, NoCI, and NoISI scenarios, respectively. The differences in the simulated concentrations between the baseline scenario and the NoELE, NoCI, and NoISI scenarios were employed to estimate the ambient $PM_{2.5}$ concentration attributable to the power, cement, and ISI sectors, respectively. The uncertainties of the emission inventory of each source were quantified with 10000 times Monte Carlo simulation based on the uncertainty range of the activity data, emission factors, and end-of-pipe control efficiencies (Supplementary Note 11)[58].

The population-weighted exposure (PWE) attributable to $PM_{2.5}$ pollution originating from each industry sector can be calculated as follow:

$$PWE = \frac{1}{P}\sum_i P_i \cdot C_{i,s} \qquad (1)$$

where $P$ indicates the total population, $P_i$ indicates the population in each geographic unit ($i$), and $C_{i,s}$ is the ambient $PM_{2.5}$ concentration attributable to each emission sector ($s$) in each geographic unit ($i$), respectively. $PM_{2.5}$-induced toxic potencies were integrated to estimate the health-relevant risk index for exposure to industrial-emitted $PM_{2.5}$, including estimation of the $PM_{2.5}$-induced OS potency- and CT potency-adjusted exposure risks ($TPAE_{OS}$ and $TPAE_{CT}$). $TPAE_{OS}$ and $TPAE_{CT}$ were evaluated as $TPAE_{OS} = C_{i,s} \times OS_{i,s}$ and $TPAE_{CT} = C_{i,s} \times CT_{i,s}$, respectively, where $OS_{i,s}$ and $CT_{i,s}$ are the toxic equivalent values of the $PM_{2.5}$-induced OS potency and CT potency, respectively, originating from each industry sector ($s$) in each geographic unit ($i$). $OS_{i,s}$ and $CT_{i,s}$ (Supplementary Note 9 and Supplementary Table 6) were estimated based on toxic units determined from the $PM_{2.5}$-induced OS and CT potency values, as detailed in our previous study[56]. The population-weighted $TPAE_{OS}$ and $TPAE_{CT}$ can be estimated as $PWE_{i,s} \times OS_{i,s}$ and $PWE_{i,s} \times CT_{i,s}$, respectively (Supplementary Table 7). The uncertainties of population-weighted TPAE were quantified based on the uncertainties of $PM_{2.5}$-induced toxic potency and $PM_{2.5}$ emission inventory (Supplementary Note 11).

## Estimation of cost reduction

The unit cost ($UCost$) of the implementation of ULE standards for $PM_{2.5}$ emission control in the ISI, power, and cement industries can be calculated as follows:

$$UCost_{i,s} = \Delta TCost_{i,s} / \Delta Emis_{i,s} \qquad (2)$$

where $i$ and $s$ denote a given end-of-pipe control measure and emission sector, respectively, $\Delta Emis$ is the abatement potential, and $TCost$ is the total cost, which can be calculated with the same method as that in our previous study[60–62], as follows:

$$TCost_{i,s} = CC_{i,s} + FOM_{i,s} + FUEL_{i,s} \qquad (3)$$

where $CC$ is the annual average capital cost, $FOM$ is the fixed operating and maintenance cost, and $FUEL$ is the fuel cost per year. $CC$ can be estimated as follows:

$$CC_{i,s} = Cost_{i,s} \times \frac{\alpha(1+\alpha)^t}{(1+\alpha)^t - 1} \qquad (4)$$

where $Cost_{i,s}$ is the initial investment cost, $\alpha$ is the discount rate, $t$ is the lifetime of the considered end-of-pipe control measure. The discount rate and lifetime of the end-of-pipe control measures in these sectors were set to 10% and 20 years, respectively, based on a comprehensive analysis of pertinent literature and government reports[62]. The unit cost of the implementation of ULE standards for toxic potency-adjusted PM emission control (including estimation of OS and CT potencies) in the ISI, power, and cement industries can be calculated as follows:

$$UCost_{OS} = \Delta TCost_{i,s}/(\Delta Emis_{i,s} \times OS_{i,s}) \quad (5)$$

$$UCost_{CT} = \Delta TCost_{i,s}/(\Delta Emis_{i,s} \times CT_{i,s}) \quad (6)$$

Further descriptions of the methods and data for unit cost calculation are provided in Supplementary Note 10 and Supplementary Tables 14–17.

## Data availability
All data supporting the conclusions in this study are provided in the manuscript, the Supplementary Materials, and the Source data file. Source data are provided with this paper.

## Code availability
The WRF and CMAQ models are open-source models. Their source codes are available on the respective release websites. (https://github.com/NCAR/WRFV3/releases; https://github.com/USEPA/CMAQ/releases).

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

## Acknowledgements

We thank Jianguo Deng from Tsinghua University, Anyuan Cheng, Lixin Zheng, Xiang Ding, Yaoqiang Huo, and Anlin Liu from Fudan University, Yangyang Guo from Institute of Process Engineering, CAS, and Weizhuo Yan from Jiangsu Environmental Protection Group Co., LTD. for supporting in field sample collection. This work was funded by the National Natural Science Foundation of China (Nos.T2122006, 22188102, U22A20405, 92143301, 92043302, and 42007393 for supporting Q.L., S.W., Q.L., Q.L., X.D.L., and L.J.), Ministry of Science and Technology of China (No.2022YFC3700501 for supporting Q.L.), New Cornerstone Science Foundation through the XPLORER PRIZE (for supporting S.W.), Samsung Advanced Institute of Technology (for supporting S.W., H.Z., B.Z., and Q.W.), China Postdoctoral Science Foundation (Nos. BX20220088, 2022M710741, and 2022M711800 for supporting D.W., D.W., and H.Z.), and the Research Grants Council of Hong Kong (Nos.15213922, 25210420, and T24-508/22-N for supporting L.J., L.J., and X.D.L.).

## Author contributions

Q. L. and S. W. conceived and designed the experiments. D. W., Q. L., Y. L. and X. C. performed the experiments. D. W., H. Z., R. L., S. L. and F. Z. analyzed the data and prepared graphs; B. Z., L. J., Q. W., T. L., J. J., L. W., X. L., J. C. and J. H. contributed materials/analysis tools. D. W., H. Z. and Q. L. wrote the manuscript, and all authors provided revisions to the manuscript.

## Competing interests

All the authors declare no competing interests.
