## [Peer Review File · Nature Communications]

Achieving health-oriented air pollution control requires integrating unequal toxicities of industrial particlesReviewers' comments:

Reviewer #1 (Remarks to the Author):

This study proposes health-oriented control strategies by integrating the unequal toxic potencies of the most polluting industrial PMs in China. The authors indicate that PM_{2.5} emissions from the iron and steel industry (ISI) exhibit about one order of magnitude higher toxic potency than those of the cement and power industries. Compared with the current mass-based control strategy, the health-oriented control strategy could generate a higher reduction in population-weighted toxic potency-adjusted PM_{2.5} exposure among polluting industries in China. The authors highlight that a health-oriented air pollution control strategy is urgently required to achieve cost-effective reductions in particulate exposure risks.

This manuscript is well-designed, organized, and presented, providing comprehensive measurements from various sources across mainland China. The flaw and data analysis are reasonable. My only concern is how robustly the authors chose oxidative potential (OP) concentrations in PM_{2.5} as an index to propose the control strategy rather than PM_{2.5} mass concentrations, which have been mainly treated as an exposure surrogate associated with adverse health outcomes in epidemiological studies. The methodology using OP to propose control measures is somewhat similar to the authors' previous study (Wu et al., 2022, Nature Energy), and it would be better to discuss the above issues. Overall, this is a well-written article and suitable for publication in this journal after minor revisions.

Specific comments:

1. Lines 70-71: Why did the authors select the iron and steel industry (ISI), power industry, and cement industry as the targets of industrial sectors in this study? What contribution do these industrial sectors make to ambient air pollution or PM_{2.5} levels?

2. Lines 172-175: The results indicate that the PM_{2.5}-induced oxidative stress (OS) potency could mainly be attributed to PM_{2.5}-bound toxic metals. Calculating PM_{2.5}-induced OS potency for selected metals and PAHs is based on reference models, which can be found in the study by Jin et al. (2019). That study provided EC_{1.5} and REP for valence metals. However, ICP-MS does not analyze metal valence. The authors should clarify the OS potency concentrations from metals and make assumptions if necessary. In addition, the calculations of OP are not transparent the authors only presented references.

Reviewer #2 (Remarks to the Author):

The manuscript titled "Achieving health-oriented air pollution control requires integrating unequal toxicities of industrial particles" highlights the need for health-oriented control strategies for PM_{2.5} pollution, which takes into account the toxic potency of the PM emitted from different industries. The author proposes a health-oriented control strategy that prioritizes controlling PM_{2.5} emissions from the iron and steel industry (ISI) due to their significantly higher toxic potency compared to those of the cement and power industries. The proposed health-oriented control strategy is shown to be more effective in reducing population-weighted toxic potency-adjusted PM_{2.5} exposure compared to the current mass-based control strategy. Additionally, the marginal abatement cost per unit of toxic potency-adjusted mass of ISI-emitted PM_{2.5} is lower than that of the other two sectors, making it a more cost-effective solution. In general, the content of the manuscript is of great significance with large sampling data and was well written. Some issues need to be revised in the manuscript before it is considered for publication in Nature communication.

1. Have any prior experimental studies investigated the toxic effects of industrial PM_{2.5}? It's necessary to compare the findings with earlier research.

2. Numerous studies in environmental epidemiology have investigated the impact of PM_{2.5}

compositions on public health. It is essential for authors to analyze and contrast the variations between their study and previous epidemiology research in this field.

3.Line 62-63 "while many of the major PM components, including inorganic sulfate and nitrate, are commonly considered innocuous species¹⁷.", the author should double check the sentence. The reference is out of date and there are many epidemiology studies found the toxic of inorganic sulfate. Modifying the acidity and alkalinity levels could potentially impact the toxicity of metals present in PM_{2.5}.

4.Line 140-142 "Here, the targeted chemicals are focused on metals and PAHs, which are commonly regarded as key toxic components associated with the biological effects of PM_{2.5}.", please explain why black carbon or elemental carbon is not a targeted chemicals in this studies?

5.It is recommended to include the proportion of certain key constituents, such as inorganic sulfate and nitrate, in Supplementary Table 4.

6.What methodology did the author employ to compute the information presented in Supplementary Figure 7a?

Reviewer #3 (Remarks to the Author):

While I agree with the authors that the toxicity of particulate matter varies with source, the sampling collection and handling methods used in this work are not appropriate for the toxicity tests they perform. The sampling methods documented in the supplementary materials fails to collect the condensible particulate matter, which is where much of the most toxic components result in the ambient air. What is required is a dilution sampling system to allow the reaction and condensation closer to what happens once the particles, vapors, and gases are released into the atmosphere. One appropriate method is described here: England GC, Watson JG, Chow JC, Zielinska B, Chang MC, Loos KR, Hidy GM. Dilution-based emissions sampling from stationary sources: Part 1--Compact sampler methodology and performance. *J Air Waste Manag Assoc.* 2007 Jan;57(1):65-78. doi: 10.1080/10473289.2007.10465291. PMID: 17269232.

A second problem with the methods is that no mention is made of post collection sample handling to protect the sample acidity from neutralization. A proper storage would place samples collected into containers with citric acid coated material to prevent neutralization of acids, such as sulfuric acid, until use in the toxicological tests, also under ammonia free conditions. This is especially important in fossil fuel combustion particulate matter, high in acidic sulfur, which is known to activate transition metals by making them bioavailable (e.g., see: Fang, T.; Guo, H.; Zeng, L.; Verma, V.; Nenes, A.; Weber, R.J. Highly Acidic Ambient Particles, Soluble Metals, and Oxidative Potential: A Link between Sulfate and Aerosol Toxicity. *Environmental Science & Technology* 2017, 51, 2611-2620, doi:10.1021/acs.est.6b06151).

The role of sulfates is dismissed as relatively non-toxic by the authors in their discussion, but this is not a full consideration of its role. In isolation, it is true that its toxicity is low, but in combination with metals (as always occurs in the real world) acidic sulfur has an important role, as indicated by the epidemiological evidence, which the authors fail to acknowledge. For example, see: Weichenthal S, Lavigne E, raub A, Umbrio D, You H, Pollitt K, Shin T, Kulka R, Stieb DM, Korsiak J, Jessiman B, JBrook JR, Hatzopoulou M, Evans G, Burnett RT, Association of Sulfur, Transition Metals, and the Oxidative Potential of Outdoor PM_{2.5} with Acute Cardiovascular Events: A Case-Crossover Study of Canadian Adults. *Environmental Health Perspectives.* *Environ Health Perspect* 129(10). (2021). doi: 30 10.1289/EHP9449.

Overall, it is unfortunate that the stack sampling and sample handling Methodology is so flawed, as the type of toxicological tests the authors performed might have provided much needed new insights into the relative toxicities of differing particles.

Reviewer #1**Comments:**

This study proposes health-oriented control strategies by integrating the unequal toxic potencies of the most polluting industrial PMs in China. The authors indicate that PM_{2.5} emissions from the iron and steel industry (ISI) exhibit about one order of magnitude higher toxic potency than those of the cement and power industries. Compared with the current mass-based control strategy, the health-oriented control strategy could generate a higher reduction in population-weighted toxic potency-adjusted PM_{2.5} exposure among polluting industries in China. The authors highlight that a health-oriented air pollution control strategy is urgently required to achieve cost-effective reductions in particulate exposure risks.

This manuscript is well-designed, organized, and presented, providing comprehensive measurements from various sources across mainland China. The flaw and data analysis are reasonable. My only concern is how robustly the authors chose oxidative potential (OP) concentrations in PM_{2.5} as an index to propose the control strategy rather than PM_{2.5} mass concentrations, which have been mainly treated as an exposure surrogate associated with adverse health outcomes in epidemiological studies. The methodology using OP to propose control measures is somewhat similar to the authors' previous study (Wu et al., 2022, Nature Energy), and it would be better to discuss the above issues. Overall, this is a well-written article and suitable for publication in this journal after minor revisions.

Response:

We appreciate the reviewer for pointing out the potential impact of our study, as well as many valuable suggestions for improving our manuscript. All the comments and suggestions have been carefully addressed in the revised version of the manuscript, while the point-to-point response to the detailed comments is provided as below.

Following the suggestion, the related description was added in the revised manuscript as: “Oxidative stress is established as one of the important mediators leading to adverse health outcomes of air pollution. Therefore, oxidative potential is a good and robust starting point to compare the mass-based and health-oriented frameworks in both technical and economic senses¹. Furthermore, increasing number of epidemiological studies have found OP was more strongly associated with cardiorespiratory disease than PM_{2.5} mass^{2,3}, indicating that OP may be a more health relevant health metric than PM_{2.5} mass. In addition, the such efforts should be extended to other health-relevant endpoints such as inflammatory responses and DNA damage. This study is distinguished from our previous work in that we provide a comprehensive econometric analysis to optimize the path to health-oriented air pollution control among different industry sectors beyond the population exposure risk estimates which we ended up with in the previous work.”

Specific comments:

1.1 *Lines 70-71: Why did the authors select the iron and steel industry (ISI), power industry, and cement industry as the targets of industrial sectors in this study? What contribution do these industrial sectors make to ambient air pollution or PM_{2.5} levels?*

Response:

Following the suggestion, the related description was added in the revised ‘Supplementary Materials’ as: “Iron and steel industry (ISI), power industry, and cement industry were chosen as the target industries for this study due to their significant contributions to ambient PM_{2.5} pollution worldwide. These three sectors were the top polluting sectors among all industry sources, accounting for 15.1% of global total anthropogenic PM_{2.5} emissions⁴. The source apportionment of ambient PM_{2.5} in China from 2005 to 2015 also suggested that power industry and industrial processes (mainly consist of ISI and cement industry) contributed 34%, 32%, and 24% in 2005, 2010, and 2015, respectively^{5,6}. Their contributions could be even higher in key regions, such as Beijing-Tianjin-Hebei region, Yangtze River Delta, and Pearl River Delta. Therefore, analyzing the environmental impact of these sectors is essential to better design emission control policies in China, particularly from the perspective of toxic potency.”

1.2 Lines 172-175: *The results indicate that the PM_{2.5}-induced oxidative stress (OS) potency could mainly be attributed to PM_{2.5}-bound toxic metals. Calculating PM_{2.5}-induced OS potency for selected metals and PAHs is based on reference models, which can be found in the study by Jin et al. (2019). That study provided EC_{1.5} and REP for valence metals. However, ICP-MS does not analyze metal valence. The authors should clarify the OS potency concentrations from metals and make assumptions if necessary. In addition, the calculations of OP are not transparent the authors only presented references.*

Response:

We thank the reviewer for this insightful comment. Metals are often combined with oxygen as metallic oxide and majority of them are usually in the form of the highest valence in PM_{2.5}. For example, iron in ambient PM_{2.5} from the San Joaquin Valley, CA was mostly as Fe (III) oxides⁷. The majority (>90%) of arsenic is in the form of As (V) in PM_{2.5} in two urban clusters of China (the Yangtze River Delta Region and Pearl River Delta Region)⁸. Hence, we assume that all the toxic metals contained in the PM exist in the highest valence during estimating the metal contributions to PM toxicities.

The related description was added in the revised manuscript as: “The PM_{2.5}-induced OS potency contributed from metals were estimated basing on the tested chemicals performed in the highest valence due to majority of them are usually in the form of the highest valence in PM_{2.5}^{7,8}.” to clarify the OS potency concentrations contributed from metals.

The calculations of PM_{2.5}-induced oxidative stress (OS) potency was added in the revised ‘Supplementary Materials’ as follow:

$$\text{IR} = \frac{\text{Fluorescence intensity (sample)}}{\text{Fluorescence intensity (control)}}$$

$$\text{IR} = 1 + \text{slope} \times \text{concentration},$$

$$\text{EC}_{1.5} = \frac{0.5}{\text{slope}},$$

where IR indicate the ROS induction ratio of the sample relative to the control. Linear concentration-effect curves with an intercept of 1 and a fitted slope were used to determine the effect concentration at a ROS induction ratio of 1.5 (EC_{1.5}).

Reviewer #2

Comments:

The manuscript titled “Achieving health-oriented air pollution control requires integrating unequal toxicities of industrial particles” highlights the need for health-oriented control strategies for PM_{2.5} pollution, which takes into account the toxic potency of the PM emitted from different industries. The author proposes a health-oriented control strategy that prioritizes controlling PM_{2.5} emissions from the iron and steel industry (ISI) due to their significantly higher toxic potency compared to those of the cement and power industries. The proposed health-oriented control strategy is shown to be more effective in reducing population-weighted toxic potency-adjusted PM_{2.5} exposure compared to the current mass-based control strategy. Additionally, the marginal abatement cost per unit of toxic potency-adjusted mass of ISI-emitted PM_{2.5} is lower than that of the other two sectors, making it a more cost-effective solution. In general, the content of the manuscript is of great significance with large sampling data and was well written. Some issues need to be revised in the manuscript before it is considered for publication in Nature communication.

Response:

We appreciate the reviewer for pointing out the significant impact of this study. All the concerns and suggestions from the reviewer were carefully addressed as below.

2.1 *Have any prior experimental studies investigated the toxic effects of industrial PM_{2.5}? It's necessary to compare the findings with earlier research.*

Response:

We thank the reviewer for this insightful comment. So far as we know, no prior experimental study systematically investigates and quantifies the toxic effects of industrial PM_{2.5} so that we cannot make quantitative comparison. The related description was added in the revised manuscript as: “Future work to quantify and rank the toxicity of individual industry source-emitted PM_{2.5} are strongly recommended for further quantitative cost-benefit analysis and finally to formulate cost-effective control strategies to effectively mitigate source-specific PM_{2.5} effects on public health.”

2.2 *Numerous studies in environmental epidemiology have investigated the impact of PM_{2.5} compositions on public health. It is essential for authors to analyze and contrast the variations between their study and previous epidemiology research in this field.*

Response:

Following the suggestion, the related description was added in the revised manuscript as: “The toxicity results observed from industrial PM are consistent with epidemiological studies, which indicate transition metals are likely to be responsible for the adverse effects of PM, including induce oxidative stress, and impairment in lung function^{9, 10}. Furthermore, multiple epidemiological evidence also shows the association between industrial PM emissions and adverse health outcomes¹¹⁻¹³. On the global scale, industrial sector emissions roughly account for 11.7% of the PM_{2.5} disease burden¹¹.”

2.3 *Line 62-63 “while many of the major PM components, including inorganic sulfate and nitrate, are commonly considered innocuous species¹⁷.”, the author should double check*

the sentence. The reference is out of date and there are many epidemiology studies found the toxic of inorganic sulfate. Modifying the acidity and alkalinity levels could potentially impact the toxicity of metals present in PM_{2.5}.

Response:

Following the suggestion, the related clarifications were added in the revised ‘Introduction’ section as: “while many of the major PM components, including inorganic sulfate and nitrate, might have interactive effects with the known toxic species.”

2.4 *Line 140-142 “Here, the targeted chemicals are focused on metals and PAHs, which are commonly regarded as key toxic components associated with the biological effects of PM_{2.5}.”, please explain why black carbon or elemental carbon is not a targeted chemicals in this studies?*

Response:

The chemical composition (i.e., organic carbon, elemental carbon (EC), water soluble inorganic ions, and elements) of PM_{2.5} emitted from three industrial sectors were determined based on standard analysis protocol, as shown in *Supplementary Fig.4* and *Supplementary Table 4*. Since the EC (i.e., char-EC and soot-EC) only accounts for minor fraction (less than 1%) of total PM_{2.5} mass, and the oxidative stress potency and cytotoxicity potency of char-EC and soot-EC are two magnitude lower than that of identified PAHs and metals. The related description was added in the revised manuscript as: “The role of EC was not taken into consideration due to its minor mass concentration in per unit PM_{2.5} mass and the relative lower toxic potency compared to PAHs and metals.”

2.5 *It is recommended to include the proportion of certain key constituents, such as inorganic sulfate and nitrate, in Supplementary Table 4.*

Response:

Following the reviewer’s suggestion, the proportion of certain inorganic constituents (i.e., sulfate, and nitrate) was added in the revised ‘Supplementary Table 4’ as:

Industrial sector	OC (%)	EC (%)	WSI (%)	Sulfate (%)	Nitrate (%)	Element (%)	EC _{1.5} (µg/mL)	IC ₂₀ (µg/mL)
Iron and steel	5.71 ± 1.33	0.90 ± 0.82	30.8 ± 14.1	9.71 ± 8.34	0.15 ± 0.15	50.1 ± 9.2	19.1 ± 10.9	46.4 ± 28.2
Power	2.33 ± 2.16	0.84 ± 0.69	49.1 ± 15.9	12.6 ± 11.2	0.88 ± 1.19	33.6 ± 16.2	94.9 ± 41.9	561 ± 226
Cement	1.74 ± 1.68	0.34 ± 0.48	38.3 ± 7.1	8.03 ± 3.91	10.7 ± 2.1	47.9 ± 9.0	103.6 ± 46.3	442 ± 221

2.6 *What methodology did the author employ to compute the information presented in Supplementary Figure 7a?*

Response:

The contribution of tested metals and PAHs to overall PM_{2.5}-induced oxidative stress potency was estimated based on concentration additional model. The detailed estimation method of percent contribution was added in the revised ‘Supplementary Materials’ as follow: “Percent concentration = BEQ_{chem}/BEQ_{PM}, BEQ_{chem} = $\sum_{i=1}^n (C_i \text{REP}_i)$, REP_i = EC_{1.5, t-BHQ}/EC_{1.5, i}, BEQ_{PM} = EC_{1.5, t-BHQ}/EC_{1.5, PM}, where BEQ_{chem} is the equivalent concentration of the each identified compound, BEQ_{PM} is the equivalent concentration of PM sample extract estimated against that of t-BHQ as the reference compound, REP_i is the relative effect potency of each identified chemical (i) for ROS generation estimated against that of the reference compound t-BHQ.”

Reviewer 3#

Comments:

While I agree with the authors that the toxicity of particulate matter varies with source, the sampling collection and handling methods used in this work are not appropriate for the toxicity tests they perform.

...

Overall, it is unfortunate that the stack sampling and sample handling Methodology is so flawed, as the type of toxicological tests the authors performed might have provided much needed new insights into the relative toxicities of differing particles.

Response:

We appreciate the reviewer for pointing out the novelty of this study and thank the reviewer for these valuable comments. We are sorry for no detailed description of handling protocol methods and the related discussion for the selected in-stack sampling method in the manuscript. All the commonly used international standard methods and for stationary emission sources and the advantages and limitations of each method were carefully reviewed and summarized. While there are some situations limit the applicability of dilution sampling in this study. The theoretical calculations of particle acidity indicate industrial particles are alkaline regardless of the employed sampling methods. The related discussion was added in the revised manuscript and Supplementary Materials, as detailed in *response 3.1* to *response 3.3*. We believe such additional information can address the reviewer's concerns on the sampling method.

3.1 *The sampling methods documented in the supplementary materials fails to collect the condensible particulate matter, which is where much of the most toxic components result in the ambient air. What is required is a dilution sampling system to allow the reaction and condensation closer to what happens once the particles, vapors, and gases are released into the atmosphere. One appropriate method is described here: England GC, Watson JG, Chow JC, Zielinska B, Chang MC, Loos KR, Hidy GM. Dilution-based emissions sampling from stationary sources: Part 1--Compact sampler methodology and performance. J Air Waste Manag Assoc. 2007 Jan;57(1):65-78. doi: 10.1080/10473289.2007.10465291. PMID: 17269232.*

Response:

We thank the reviewer for this comment on the sampling method. We carefully addressed this issue and made revision in the following three aspects.

(1) The in-stack sampling and the dilution sampling are both widely used methods for sampling/monitoring PM_{2.5} emissions from stationary sources worldwide. Each method possesses both advantages and limitations. While in-stack sampling is a practical method to obtain abundant PM samples (in mass) at low PM concentration in stack for post cellular-based toxicity tests.

Related description was added in the revised Manuscript as “Based on reviewing the commonly used sampling method for industrial sources and comparing the advantages and limitations of each method (Supplementary Note 5 and Supplementary Tables 19–20), ...” and in the Supplementary Materials as:

Supplementary Table 19: Summary of international standard method for determination of PM emissions from stationary sources

Sampling method	Standard method	Extraction method	Applicability
In-stack sampling method	GB 5468-91	Isokinetic sampling	Measurement of smoke and dust emission from boilers
	GB/T 16157-1996	Isokinetic sampling	Determination of particulates and sampling methods of gaseous pollutants from exhaust gas of stationary source
	HJ/T397-2007	Isokinetic sampling	Technical specifications for emission monitoring of stationary source
	ISO 23210:2009	Two-stage impactor	Determination of PM ₁₀ and PM _{2.5} mass concentrations at stationary emission sources.
	EPA Method 5	Isokinetic sampling	Determination of PM emissions from stationary sources
	EPA Method 17	Isokinetic sampling	Determination of PM emissions from stationary sources
	EPA Method 201A	Cyclone samplers	Determination of PM, PM ₁₀ , PM _{2.5} emissions from stationary sources (constant sampling rate procedures)
EPA Method 202	Dry impinger	Determination of condensable PM emissions from stationary sources	
Dilution sampling method	ISO 25597:2013	Dilution sampling system	Determining PM _{2.5} and PM ₁₀ mass in stack gases using cyclone samplers and sample dilution
	EPA Method CTM-039	Dilution sampling system	Measurement of PM _{2.5} and PM ₁₀ emissions by dilution sampling (constant sampling rate procedures)

“The in-stack sampling method (International Organization for Standardization (ISO) 23210:2009¹⁴, Environmental Protection Agency (EPA) Method 5¹⁵, Method 17¹⁶, Method 201A¹⁷, Method 202¹⁸) and dilution sampling method (ISO 25597:2013¹⁹, EPA CTM 039²⁰) are both promulgated and recommended by United States EPA and ISO for determination of PM emissions from stationary sources (Supplementary Table 19). While the in-stack sampling is the nationwide standard method for determination of PM emissions from stationary sources in China (GB 548-91²¹, GB/T 16157-1996²², and HJ/T397-2007²³). The in-stack sampling was widely employed for determination of PM emissions from ultralow coal-fired power plants^{24, 25}, iron and steel industry^{26, 27}, and cement plants²⁸ in previous studies.

We carefully compared the two methods, and the advantages and limitations of each method as well as its applicability of *in vitro* cellular-based toxicity tests are listed in Supplementary Table 20. Dilution sampling method can achieve a rapid condensation and cooling process, such as employed in measurement of PM_{2.5} emitted from residential combustion and transportation presented in our previous studies. While it is recognized that there are some situations that limit the applicability of the dilution sampling methods in

stationary sources¹⁹. No standard dilution parameter (i.e., dilution ratio, nominal residence time, mixing section Reynolds number) lead to large uncertainties during sampling processes and post analysis²⁹. In order to sampling enough PM_{2.5} mass for chemical analysis, each dilution sampling needs to be performed for at least 1-2 days at the monitoring platform of stacks due to low PM_{2.5} concentration at stack under current control policy³⁰. Since dilution sampling prefer to perform at stable conditions¹⁹. Long-time sampling process can lead to additional uncertainties due to the uncontrollable variations in stack (i.e., the velocity, temperature, and concentration of flue gas fluctuate widely in stationary sources)¹⁹. Furthermore, the PM losses and filter broken during the long-time dilution process can result in large uncertainties for filter weighing and other analysis³¹. In addition, the presence of particles in the dilution air may lead to high measurement background¹⁹. For applicability of *in vitro* cellular-based toxicity tests, the dilution sampling is unable to capture adequate PM mass to support post analysis due to the very low concentrations at stack under the stringent emission control standards, which anticipated sufficient PM_{2.5} mass for reliable and replicable results³². In contrast, in-stack sampling method can achieve isokinetic stack sampling and obtain adequate representative PM_{2.5} samples at a relative stable condition. The sulfur trioxide, an important precursor of sulfate and condensable PM, can be easily condensed and transformed during in-stack sampling process. Hence, the in-stack sampling method is the most practical way to collect PM_{2.5} samples from stationary sources for *in vitro* tests and reliable biological results in this study.”

Supplementary Table 20: Advantages, limitations, and applicability of *in vitro* cellular-based tests faced with in-stack sampling and dilution sampling method.

Parameter	In-stack sampling method	Dilution sampling method
Advantages	 Can achieve isokinetic stack sampling Can obtain representative samples in a short sampling time Feasible to operate and maintenance Can achieve sufficient PM mass at stable in-stack condition 	 Can achieve a rapid condensation and cooling process
Limitations	 Unable to mimic a rapid condensation process 	 No standard dilution parameter (dilution ratio, residence time, etc.) lead to large uncertainties; Long-time sampling processes lead to additional uncertainties due to uncontrollable variations of flue gas in stacks (i.e., velocity, temperature, and concentration); Higher losses of particles during sampling process due to particle deposition; High measurement background due to the presence of particles in the dilution air; Not applicable to measure emissions from stacks that have entrained moisture droplets; Not applicable to measure emissions from stacks that have fluctuations in velocity, temperature, and concentration; Unable to collect sufficient PM mass for toxicity tests
Applicability of cellular-based toxicity tests	The most practical way	Not suitable

(2) Current industrial PM control policies worldwide focus on targeting the reduction in PM mass emissions (namely, the in-stack PM mass concentrations), while condensable PM is not included. Additionally, modeling studies focusing on health effects induced by industrial PM_{2.5} pollution have also used in-stack measurement data³³. Hence, the in-stack sampling method is suitable to estimate the effectiveness of current mass-based PM control strategy on mitigating regional PM-induced health impacts. Future work is indeed required to quantify the health impacts of condensable PMs released by industrial sources. The related description was added in the revised ‘Supplementary Materials’ as:

“In addition, clean air actions taken in industrial sectors worldwide have enforced a series of control measures (i.e., install the continuous emission monitoring system, implement advanced control technologies) to target the reduction in PM mass emissions (namely, the in-stack PM mass concentrations) while condensable PM has not been included³⁴⁻³⁶. Furthermore, modeling studies focusing on health effects induced by industrial PM_{2.5} pollution have used in-stack measurement data³⁷. Hence, the in-stack sampling method is more suitable to estimate the effectiveness of current mass-based PM control strategy on mitigating regional PM-induced health impacts.”

(3) The concerned chemical composition (i.e., sulfate) of PM_{2.5} samples observed via the in-stack sampling method are consistent with those obtained via the dilution sampling method presented in recent study under similar stack condition³⁸. Furthermore, the theoretical calculations of H⁺ levels³⁹ indicate the industrial PMs obtained from the two sampling methods are all alkaline (*Supplementary Table 21*). The results suggest that the solubility of metals emitted from industrial sources is unlikely to be altered, since only highly acidic particles (pH < 2) may change their solubilities⁴⁰. There is no evidence quantifying the effect of pH on industrial PM toxicity⁴¹. More future works are needed to address these concerns. The related discussion was added in the revised manuscript as:

“The chemical composition (i.e., sulfate) of PM_{2.5} samples (in mass fraction) observed from in-stack sampling method are consistent with those obtained from dilution sampling method presented in previous studies (*Supplementary Table 3*)³⁰. The theoretical calculations of [H⁺] value (*Supplementary Table 21*) indicate the industrial particles are all alkaline regardless of the employed sampling method³⁸. Future works are needed to quantification the effect of pH on industrial PM toxicity⁴¹.”

Supplementary Table 21: Comparison of PM chemical component of sulfate and the proton loading (H⁺) in industrial PM observed from in-stack sampling and dilution sampling methods.

Source	Sampling method	Sampling site	SO ₄ (%)	H ⁺	Reference
Iron and steel plant (sintering)	Dilution sampling	Stack	3.40 ± 1.63	-0.16	Zeng et al., 2021 ³⁸
Ultra-low emission power plant		Stack	23.2 ± 1.28	-4.54	
Cement plant		Stack	6.52 ± 0.88	-1.65	
Iron and steel plant (sintering)	In-stack sampling	Stack	6.85 ± 1.01	-3.16	This study
Ultra-low emission power plant		Stack	25.4 ± 11.9	-17.6	
Cement plant		Stack	8.03 ± 3.91	-0.60	

The calculation of particle acidity was added in the revised ‘Supplementary Materials’ as:
“The acidity of industrial particles obtained from in-stack sampling and dilution sampling was estimated based on the ion balance method, which is one of the most commonly used methods to estimate the proton loading (H^+) in particles³⁹. The H^+ is estimate as $[H^+] = \sum n_i[anion_i] - \sum n_i[cation_i]$, where n_i is the charge of species i , $[anion_i]$ and $[cation_i]$ are the molar concentrations of anion and cation species, respectively. If the $[H^+]$ value is less than zero, then the difference is attributed to OH^- .”

3.2 *A second problem with the methods is that no mention is made of post collection sample handing to protect the sample acidity from neutralization. A proper storage would place samples collected into containers with citric acid coated material to prevent neutralization of acids, such as sulfuric acid, until use in the toxicological tests, also under ammonia free conditions. This is especially important in fossil fuel combustion particulate matter, high in acidic sulfur, which is known to activate transition metals by making them bioavailable (e.g., see: Fang, T.; Guo, H.; Zeng, L.; Verma, V.; Nenes, A.; Weber, R.J. Highly Acidic Ambient Particles, Soluble Metals, and Oxidative Potential: A Link between Sulfate and Aerosol Toxicity. Environmental Science & Technology 2017, 51, 2611-2620, doi:10.1021/acs.est.6b06151).*

Response:

We are sorry for too brief description of sample handling methods in the manuscript, although we mentioned that ‘*The field sampling and quality control methods for the industrial emissions have been detailed in our previous studies*’ with provided references. Following this suggestion, the additional description of sample handling method was added in the revised manuscript as: “After sampling, each $PM_{2.5}$ sample was placed into a petri dish and sealed with a zipper bag and stored at $-20^\circ C$ immediately before undergoing analysis”.

The bioavailability of metals emission from industrial sources may not increase in this study. Since the theoretical calculations of H^+ levels³⁹ indicate the industrial particles obtained from both in-stack sampling and dilution sampling method are alkaline. While only highly acidic particles ($pH < 2$) may change the solubility of metals⁴⁰, as detailed in the reference mentioned by the review. The related description is detailed in *response 3.1-(3)*.

3.3 *The role of sulfates is dismissed as relatively non-toxic by the authors in their discussion, but this is not a full consideration of its role. In isolation, it is true that its toxicity is low, but in combination with metals (as always occurs in the real world) acidic sulfur has an important role, as indicated by the epidemiological evidence, which the authors fail to acknowledge. For example, see: Weichenthal S, Lavigne E, raub A, Umbrio D, You H, Pollitt K, Shin T, Kulka R, Stieb DM, Korsiak J, Jessiman B, JBrook JR, Hatzopoulou M, Evans G, Burnett RT, Association of Sulfur, Transition Metals, and the Oxidative Potential of Outdoor $PM_{2.5}$ with Acute Cardiovascular Events: A Case-Crossover Study of Canadian Adults. Environmental Health Perspectives. Environ Health Perspect 129(10). (2021). doi: 30 10.1289/EHP9449.*

Response:

The theoretical calculations of H^+ levels³⁹ indicate the industrial particles obtained from both in-stack sampling and dilution sampling method are alkaline (Supplementary Table 21). The

results suggest that sulfate may not play vital role in altering the bioavailability of metals emission from industrial sources since only highly acidic particles (pH < 2) may change the solubility of metals⁴⁰. Detailed description is shown in **response 3.1 (3)**. Furthermore, the role of sulfate and transition metals in shaping PM toxicity is still unclear. There is no toxicological evidence that quantification the interactive effects of sulfate and transition metals. Some related discussion was added in the revised manuscript as:

“Toxicological studies (*in vitro* and *in vivo* tests) are urgently needed to quantify the interactive effects of individual chemical, such as sulfate and transition metals, and to clarify and validate the associations by integrating with epidemiological studies.”

Reference

1. Daellenbach, K. R., et al. (2020). Sources of particulate-matter air pollution and its oxidative potential in Europe. *Nature* 587, 414-419.
2. Abrams, J. Y., et al. (2017). Associations between Ambient Fine Particulate Oxidative Potential and Cardiorespiratory Emergency Department Visits. *Environ. Health Perspect.* 125, 107008.
3. Bates, J. T., et al. (2015). Reactive Oxygen Species Generation Linked to Sources of Atmospheric Particulate Matter and Cardiorespiratory Effects. *Environ. Sci. Technol.* 49, 13605-13612.
4. Huang, Y., et al. (2014). Quantification of Global Primary Emissions of PM_{2.5}, PM₁₀, and TSP from Combustion and Industrial Process Sources. *Environ. Sci. Technol.* 48, 13834-13843.
5. Zhu, Y., et al. (2018). Sources of particulate matter in China: Insights from source apportionment studies published in 1987–2017. *Environ. Int.* 115, 343-357.
6. Zheng, H., et al. (2019). Transition in source contributions of PM_{2.5} exposure and associated premature mortality in China during 2005–2015. *Environ. Int.* 132, 105111-105124.
7. O’Day, P. A., et al. (2022). Iron Speciation in Respirable Particulate Matter and Implications for Human Health. *Environ. Sci. Technol.* 56, 7006-7016.
8. Xie, J., et al. (2020). Health risk-oriented source apportionment of PM_{2.5}-associated trace metals. *Environ. Pollut.* 262, 114655.
9. Wu, S., et al. (2013). Chemical constituents of fine particulate air pollution and pulmonary function in healthy adults: The Healthy Volunteer Natural Relocation study. *J. Hazard. Mater.* 260, 183-191.
10. Liu, L., et al. (2018). Metals and oxidative potential in urban particulate matter influence systemic inflammatory and neural biomarkers: A controlled exposure study. *Environ. Int.* 121, 1331-1340.
11. McDuffie, E. E., et al. (2021). Source sector and fuel contributions to ambient PM_{2.5} and attributable mortality across multiple spatial scales. *Nat. Commun.* 12, 3594.
12. Pond, Z. A., et al. (2022). Cardiopulmonary Mortality and Fine Particulate Air Pollution by Species and Source in a National U.S. Cohort. *Environ. Sci. Technol.* 56, 7214-7223.
13. Thakrar, S. K., et al. (2020). Reducing Mortality from Air Pollution in the United States by Targeting Specific Emission Sources. *Environ. Sci. Technol. Lett.* 7, 639-645.
14. International Organization for Standardization (ISO). ISO 23210: 2009. Stationary source emissions — Determination of PM₁₀/PM_{2.5} mass concentration in flue gas — Measurement at low concentrations by use of impactors. In International Organization for Standardization: Geneva, Switzerland: 2009.
15. United States Environmental Protection Agency (US EPA). Method 5- Determination of PM

emissions from stationary sources. In US EPA: Washington, D.C.: 2020.

16. United States Environmental Protection Agency (US EPA). Method 17- Determination of PM emissions from stationary sources. In US EPA: Washington, D.C.: 2017.

17. United States Environmental Protection Agency (US EPA). Method 201A- Determination of PM10 and PM2.5 emissions from stationary sources. In US EPA: Washington, D.C.: 2010.

18. United States Environmental Protection Agency (US EPA). Method 202-Dry impinger method for determining condensable particulate emissions from stationary sources. In US EPA: Washington, D.C.: 2010.

19. International Organization for Standardization (ISO). ISO 25597: 2013. Stationary source emissions-Test method for determining PM2.5 and PM10 mass in stack gases using cyclone samplers and sample dilution. In International Organization for Standardization: Geneva, Switzerland: 2013.

20. United States Environmental Protection Agency (US EPA). Conditional Test Method 039-Measurement of PM2.5 and PM10 emissions by dilution sampling (constant sampling rate procedures). In US EPA: Washington, D.C.: 2004.

21. Measurement of smoke and dust emission from boilers. In China Environmental Press: 1991.

22. The determination of particulates and sampling methods of gaseous pollutants from exhaust gas of stationary source (In Chinese). In China Environmental Press 1996.

23. Technical specifications for emission monitoring of stationary source (In Chinese). In China Environmental Press: 2007.

24. Wu, B., et al. (2018). Effects of Wet Flue Gas Desulfurization and Wet Electrostatic Precipitators on Emission Characteristics of Particulate Matter and Its Ionic Compositions from Four 300 MW Level Ultralow Coal-Fired Power Plants. *Environ. Sci. Technol.* 52, 14015-14026.

25. Liu, W., et al. (2020). Migration and Emission Characteristics of Ammonia/Ammonium through Flue Gas Cleaning Devices in Coal-Fired Power Plants of China. *Environ. Sci. Technol.* 54, 390-399.

26. Tsai, J.-H., et al. (2007). Chemical constituents in particulate emissions from an integrated iron and steel facility. *J. Hazard. Mater.* 147, 111-119.

27. Guo, Y., et al. (2017). Chemical profiles of PM emitted from the iron and steel industry in northern China. *Atmos. Environ.* 150, 187-197.

28. Guo, Z., et al. (2021). Field measurements on emission characteristics, chemical profiles, and emission factors of size-segregated PM from cement plants in China. *Sci. Total Environ.* 818, 151822.

29. Zheng, S., et al. (2022). Impact of Dilution Ratio and Burning Conditions on the Number Size Distribution and Size-Dependent Mixing State of Primary Particles from Domestic Solid Fuel Burning. *Environ. Sci. Technol. Lett.* 9, 611-617.

30. Zeng, X., et al. (2021). Source profiles and emission factors of organic and inorganic species in fine particles emitted from the ultra-low emission power plant and typical industries. *Sci. Total Environ.* 789, 147966.

31. Hildemann, L. M., et al. (1989). A Dilution Stack Sampler for Collection of Organic Aerosol Emissions: Design, Characterization and Field Tests. *Aerosol Sci. Tech.* 10, 193-204.

32. Hill, W., et al. (2023). Lung adenocarcinoma promotion by air pollutants. *Nature* 616, 159-167.

33. Li, M., et al. (2022). Impacts of condensable particulate matter on atmospheric organic aerosols and fine particulate matter (PM2.5) in China. *Atmos. Chem. Phys.* 22, 11845-11866.

34. Crippa, M., et al. (2016). Forty years of improvements in European air quality: regional policy-industry interactions with global impacts. *Atmos. Chem. Phys.* 16, 3825-3841.

35. Shapiro, J. S.; Walker, R. (2018). Why Is Pollution from US Manufacturing Declining? The Roles of Environmental Regulation, Productivity, and Trade. *Am. Econ. Rev.* 108, 3814-54.
36. Zheng, B., et al. (2018). Trends in China's anthropogenic emissions since 2010 as the consequence of clean air actions. *Atmos. Chem. Phys.* 18, 14095-14111.
37. Lelieveld, J., et al. (2015). The contribution of outdoor air pollution sources to premature mortality on a global scale. *Nature* 525, 367-71.
38. Zeng, X., et al. (2021). Source profiles and emission factors of organic and inorganic species in fine particles emitted from the ultra-low emission power plant and typical industries. *Sci. Total Environ.* 789, 147966-147978.
39. Hennigan, C. J., et al. (2015). A critical evaluation of proxy methods used to estimate the acidity of atmospheric particles. *Atmos. Chem. Phys.* 15, 2775-2790.
40. Fang, T., et al. (2017). Highly Acidic Ambient Particles, Soluble Metals, and Oxidative Potential: A Link between Sulfate and Aerosol Toxicity. *Environ. Sci. Technol.* 51, 2611-2620.
41. Gu, B., et al. (2022). Particle toxicity's role in air pollution-Response. *Science* 375, 506-507.

Reviewers' comments:

Reviewer #1 (Remarks to the Author):

The authors have promptly responded to all my comments and questions. I suggest accepting this article.

Reviewer #2 (Remarks to the Author):

My comments are fully addressed. The manuscript can be accepted.

Reviewer #3 (Remarks to the Author):

The authors response about dilution sampling and the condensable PM_{2.5} missing from their sampling is not satisfactory. By only including the filterable PM_{2.5}, and not the condensable PM_{2.5}, they missed the majority of many of the important metals they consider. For example, see Table 3 of:

Characteristics of filterable and condensable particulate matter emitted from two waste incineration power plants in China

Gang Wang, Jianguo Deng, Zizhen Maa, Jiming Hao^{a,b}, Jingkun Jianga^{*,b}.

Science of The Total Environment

Volume 639, 15 October 2018, Pages 695-704

<https://www.sciencedirect.com/science/article/pii/S0048969718317406>

Key elements like Cu are predominantly in the Condensable PM_{2.5}, not the filterable PM_{2.5} collected by the authors.

This is especially a problem for the high temperature processes, like power plants, where losses of the condensable metals would be extremely problematic. Basically, the authors must admit that their results and conclusions could change completely if they had collected and considered the condensable metals.

Reviewer #3

Comments:

The authors response about dilution sampling and the condensable PM_{2.5} missing from their sampling is not satisfactory. By only including the filterable PM_{2.5}, and not the condensable PM_{2.5}, they missed the majority of many of the important metals they consider. For example, see Table 3 of: Characteristics of filterable and condensable particulate matter emitted from two waste incineration power plants in China

Gang Wang, Jianguo Deng, Zizhen Maa, Jiming Hao^{a,b}, Jingkun Jianga^{a,b}.

Science of The Total Environment Volume 639, 15 October 2018, Pages 695-704

<https://www.sciencedirect.com/science/article/pii/S0048969718317406>

Key elements like Cu are predominantly in the Condensable PM_{2.5}, not the filterable PM_{2.5} collected by the authors.

This is especially a problem for the high temperature processes, like power plants, where losses of the condensable metals would be extremely problematic. Basically, the authors must admit that their results and conclusions could change completely if they had collected and considered the condensable metals.

Response:

We appreciate the reviewer for re-review our manuscript. Upon careful considerations of the reviewer's comments, we have found that the reviewer did not provide scientific evidence or specific comments responding to our previously submitted responses. The main technical concern newly raised by reviewer pertains to the 'condensable metals'. We believe that the reviewer's concern is a factual error as it violates the phase transition law of metals, an incredibly important area of physics in university courses worldwide.

The related description was added to clarify the 'condensable metals' for avoiding misunderstanding in the revised 'Supplementary Material' in the subsection of 'Chemicals and Toxicity Analysis and Quality Control' as:

"The majority of metals cannot exist in the gaseous phase under the in-stack sampling temperature (i.e, 120 °C; ISO 23210:2009), which is significantly lower than the gasification temperature of the targeted 10 toxic metals (i.e., V, 3380 °C; Cr, 2761 °C; Mn, 2861 °C; Fe, 2861 °C; Ni, 2732 °C; Cu, 2562 °C; Zn, 907 °C; As, 614 °C; Cd, 767 °C; and Pb, 1740 °C)¹⁻³. All of these targeted toxic metals mainly exist in the particulate phase and cannot significantly contribute to condensable PMs⁴."

"While previous reports that indicated high concentrations of nonvolatile metals detected in condensable PMs are likely due to the results of experimental error in field measurements⁵, originating from PM-bound metals penetrating the filterable PM filters and entering into the condensable PM sampling train⁶. Even more recent studies from the same research group do not include toxic metals in their reported condensable PMs⁷. Thus, the quality of filterable PM_{2.5} samples collected by the in-stack method was assured for further toxicity assessment."

Furthermore, related description was added in the revised manuscript in the subsection of 'Discussion and policy implication' as: *"Condensable PMs originating from industrial sectors as well as..." "...were not considered, ..."*

Reference

1. Haynes, W. M., *CRC Handbook of Chemistry and Physics, 92nd Edition*. Taylor & Francis: 2011.
2. Zhang, Y., et al. (2011). Corrected Values for Boiling Points and Enthalpies of Vaporization of Elements in Handbooks. *J. Chem. Eng. Data* 56, 328-337.
3. James, A. M.; Lord, M. P., *Macmillan's Chemical and Physical Data*. Macmillan: 1992.
4. Flagan, R. C.; Seinfeld, J. H., *Fundamentals of Air Pollution Engineering*. Dover: 2012.
5. Wang, G., et al. (2018). Characteristics of filterable and condensable particulate matter emitted from two waste incineration power plants in China. *Sci. Total Environ.* 639, 695-704.
6. Zhang, H., et al. (2023). Exploration on the source of SO₄²⁻ in Condensable Particulate Matter. *Fuel* 337, 126949.
7. Wang, G., et al. (2020). Evaluating Airborne Condensable Particulate Matter Measurement Methods in Typical Stationary Sources in China. *Environ. Sci. Technol.* 54, 1363-1371.

REVIEWER COMMENTS

Reviewer #4 (Remarks to the Author):

This study reports the unequal toxicity of PMs→ emitted from industry sectors based on large field measurements and laboratory studies. The iron and steel industry-emitted PM2.5 were revealed to be much more toxic than the cement- and power industries-emitted PM2.5. The toxicities were well explained by the provided data on chemical analysis of collected samples. The authors further proposed a health-oriented air pollution control strategy for industry sectors, indicating a prior control should be implemented on iron and steel industry, by integrating experimental data, air quality model, and cost analysis.

The informative manuscript is very interesting and important for the current control of industrial air quality management. The real-world industrial PM emission profiles and their toxicities and exposure risks will be a broad interesting to potential readers in the research fields of environmental science, economy, and public health. The science-based information provided by the manuscript is important and useful for the policy making to take actions for reducing the industry-derived exposure risk for human being. Overall, this manuscript is well-organized and has high novelty.

Considering the concerns from the referee 3#, I would like to provide my comments as following. First, the condensable PM2.5, or condensable metals mentioned by the referee belong to the category of condensable particulate matter (CPM), while this study focuses on filterable particulate matter (FPM) emissions from industry sources, which is the targeted pollutants regulated by the current industrial pollution control policy worldwide. It is suggested that the author further clarify the research object or scope in the revised manuscript.

Second, the discussion of whether the metals existed in CPM or not is still in dispute in the published papers. There are some publications providing the experimental data of condensable metals in combustion flue gases, but the concentration is very low. However, the theoretical analysis support that toxic metals are mainly present in FPM rather than in CPM. To my knowledge, no solid evidence can support that metals, such as Fe, Zn, Mn, Pb, Cu, and etc. were observed in gaseous phase of in-stack flue gases or atmosphere.

Therefore, the concern of condensable metals should not break the reliability of the data and conclusion of this manuscript. Maybe, the issue of CPM is also important, but I think it can be listed in future work. The discussion of CPM in future work is encouraged to add in the revised manuscript. In a word, I recommend the acceptance of this manuscript. In addition, other minor comments are provided for further revision:

1. Given this article highlights the toxicity of metals, it is recommended that the authors need to include the discussion of the condensation of toxic metals in typical industrial flue gases in the subsection of 'Substantial discrepancies among key toxic chemicals in PM2.5.'
2. The wet plume is a commonly phenomenon for industry sources after implementing the ultra-low emission standards in China. It's recommended to include the discussion of the toxicity of wet plume after they exhaust to atmosphere and drying by ambient air.
3. Line 93-96: "Compared to the power and cement industries, the ISI involves a series of complex processes, mainly including sintering (and/or pelletizing), blast furnace (BF) based-ironmaking, basic oxygen furnace (BOF)-based or electric arc furnaces (EAF)-based steelmaking, and steel rolling", this statement is duplicate with the description in the latter paragraph. I recommend removing it. There are several redundant descriptions in other locations, the authors need to check and remove them.
4. Line 356-360: The discussion of chemical additional model is confusing and not closely related with the results. The authors could remove and short the discussion.
5. The information provided by Table 5 in SI can be found elsewhere in many other papers. It's recommended to be removed.

6. The content in Table 21 in SI is confusing and seems to be irrelevant with this study. It also could be removed to avoid misunderstanding for potential readers.

7. Line 213-236 in supporting information: I recommend simplifying the description of the sampling method, which is an international standard sampling method and well documented in numerous published studies.

Reviewer #4

Comments:

This study reports the unequal toxicity of PMs emitted from industry sectors based on large field measurements and laboratory studies. The iron and steel industry-emitted PM_{2.5} were revealed to be much more toxic than the cement- and power industries-emitted PM_{2.5}. The toxicities were well explained by the provided data on chemical analysis of collected samples. The authors further proposed a health-oriented air pollution control strategy for industry sectors, indicating a prior control should be implemented on iron and steel industry, by integrating experimental data, air quality model, and cost analysis.

The informative manuscript is very interesting and important for the current control of industrial air quality management. The real-world industrial PM emission profiles and their toxicities and exposure risks will be a broad interesting to potential readers in the research fields of environmental science, economy, and public health. The science-based information provided by the manuscript is important and useful for the policy making to take actions for reducing the industry-derived exposure risk for human being. Overall, this manuscript is well-organized and has high novelty.

Considering the concerns from the referee 3#, I would like to provide my comments as following. First, the condensable PM_{2.5}, or condensable metals mentioned by the referee belong to the category of condensable particulate matter (CPM), while this study focuses on filterable particulate matter (FPM) emissions from industry sources, which is the targeted pollutants regulated by the current industrial pollution control policy worldwide. It is suggested that the author further clarify the research object or scope in the revised manuscript.

Second, the discussion of whether the metals existed in CPM or not is still in dispute in the published papers. There are some publications providing the experimental data of condensable metals in combustion flue gases, but the concentration is very low. However, the theoretical analysis support that toxic metals are mainly present in FPM rather than in CPM. To my knowledge, no solid evidence can support that metals, such as Fe, Zn, Mn, Pb, Cu, and etc. were observed in gaseous phase of in-stack flue gases or atmosphere.

Therefore, the concern of condensable metals should not break the reliability of the data and conclusion of this manuscript. Maybe, the issue of CPM is also important, but I think it can be listed in future work. The discussion of CPM in future work is encouraged to add in the revised manuscript.

In a word, I recommend the acceptance of this manuscript. In addition, other minor comments are provided for further revision.

Response:

We appreciate the reviewer for pointing out the significance, novelty, and potential impact of our study, as well as many valuable suggestions for improving our manuscript. All the comments and suggestions have been carefully addressed in the revised version of the manuscript, while the point-to-point response to the detailed comments is provided as below.

Following the suggestion, the research scope was clarified in the revised subsection of 'Introduction' as:

“PM_{2.5} samples (refer to the filterable PM_{2.5} in this study) were collected from these three sectors in field across 21 provinces in mainland China...”

As well as in the revised subsection of 'Field measurements' as:

“An isokinetic sampling system was employed to collect PM_{2.5} emission samples (namely, the filterable PM_{2.5}) originating from stationary sources.”

The discussion of CPM in future work was added in the revised subsection of ‘Discussion and policy implication’ as:

“In future, it is necessary to investigate the potential toxicities of industrial condensable PMs and explore the quantitative connection between their toxic potencies and the corresponding chemical composition. Furthermore, quantifying the role of condensable PMs in adjusting the toxicity of filterable PMs is crucial for a comprehensively understanding of the adverse impacts originating from industry sources.”

Other minor comments:

4.1 *Given this article highlights the toxicity of metals, it is recommended that the authors need to include the discussion of the condensation of toxic metals in typical industrial flue gases in the subsection of ‘Substantial discrepancies among key toxic chemicals in PM_{2.5}.’*

Response:

Following the suggestion, the related discussion on the condensation of toxic metals was added in the revised subsection of ‘Substantial discrepancies among key toxic chemicals in PM_{2.5}’ as:

“The targeted toxic metals of concern predominately exist in the particulate phase, not the vapor phase, in typical industrial flue gases. This is mainly due to their significantly higher gasification temperatures¹⁻³. Consequently, the condensation of these matters after being discharged from the stack can be disregarded or considered negligible.”

4.2 *The wet plume is a commonly phenomenon for industry sources after implementing the ultra-low emission standards in China. It’s recommended to include the discussion of the toxicity of wet plume after they exhaust to atmosphere and drying by ambient air.*

Response:

We thank the reviewer for this insightful comment. The related discussion about the toxicity of wet plume has been added in the revised subsection of ‘Discussion and policy implication’ as: “Wet plumes, formed when saturated wet flue gas is discharged from stack and enters to the atmosphere, are frequently observed in industrial plants⁴. The impact of wet plumes on air quality and human health has raised growing concerns in recent years. However, the toxicity of wet plumes remains unclear, emphasizing the need for systematic studies to address these issues in the future.”

4.3 *Line 93-96: “Compared to the power and cement industries, the ISI involves a series of complex processes, mainly including sintering (and/or pelletizing), blast furnace (BF) based-ironmaking, basic oxygen furnace (BOF)-based or electric arc furnaces (EAF)-based steelmaking, and steel rolling”, this statement is duplicate with the description in the latter paragraph. I recommend removing it. There are several redundant descriptions in other locations, the authors need to check and remove them.*

Response:

Following the reviewer's suggestion, the related description "*Compared to...steel rolling*" were deleted in the revised subsection of '**Unequal toxic potencies of industry-emitted PM_{2.5}**'. In addition, we carefully checked the redundant descriptions and addressed these issues in the revised manuscript and '**Supporting Materials**'.

4.4 *Line 356-360: The discussion of chemical additional model is confusing and not closely related with the results. The authors could remove and short the discussion.*

Response:

Following the suggestion, the related discussion "*Additionally, the estimation...based on chemical additional model...*" were deleted in the revised manuscript.

4.5 *The information provided by Table 5 in SI can be found elsewhere in many other papers. It's recommended to be removed.*

Response:

Following the suggestion, the '**Supplementary Table 5**' and the related description was removed from the '**Supporting Materials**'.

4.6 *The content in Table 21 in SI is confusing and seems to be irrelevant with this study. It also could be removed to avoid misunderstanding for potential readers.*

Response:

Following the reviewer's suggestion, the '**Supplementary Table 21**' and the related description was removed from the '**Supporting Materials**'.

4.7 *Line 213-236 in supporting information: I recommend simplifying the description of the sampling method, which is an international standard sampling method and well documented in numerous published studies.*

Response:

Following the reviewer's suggestion, the description of the sampling method was simplified in the revised '**Supporting Materials**' as:

"The in-stack sampling method, such as International Organization for Standardization (ISO) 23210:2009⁵, Environmental Protection Agency (EPA) Method 5⁶, Method 17⁷, Method 201A⁸, Method 202⁹, has been officially recommended by United States EPA and ISO for determining PM emissions from stationary sources (see Supplementary Table 18). Such method has been widely utilized in previous studies to determine PM emissions from various sources, including ultralow emission coal-fired power plants^{10, 11}, iron and steel industry^{12, 13}, and cement plants¹⁴. Hence, a cyclone-based in-stack sampling system (C-5000, ESC, USA) was deployed to separate and collect PM_{2.5} samples from industrial flue gases, following US EPA Method 201A and Method 17."

References

1. Haynes, W. M., *CRC Handbook of Chemistry and Physics, 92nd Edition*. Taylor & Francis: 2011.
2. Zhang, Y., et al. (2011). Corrected Values for Boiling Points and Enthalpies of Vaporization of Elements in Handbooks. *J. Chem. Eng. Data* 56, 328.
3. James, A. M.; Lord, M. P., *Macmillan's Chemical and Physical Data*. Macmillan: 1992.
4. Ding, X., et al. (2021). Direct Observation of Sulfate Explosive Growth in Wet Plumes Emitted From Typical Coal-Fired Stationary Sources. *Geophys. Res. Lett.* 48, e2020GL092071.
5. International Organization for Standardization (ISO). ISO 23210: 2009. Stationary source emissions — Determination of PM₁₀/PM_{2.5} mass concentration in flue gas — Measurement at low concentrations by use of impactors. In International Organization for Standardization: Geneva, Switzerland: 2009.
6. United States Environmental Protection Agency (US EPA). Method 5- Determination of PM emissions from stationary sources. In US EPA: Washington, D.C.: 2020.
7. United States Environmental Protection Agency (US EPA). Method 17- Determination of PM emissions from stationary sources. In US EPA: Washington, D.C.: 2017.
8. United States Environmental Protection Agency (US EPA). Method 201A- Determination of PM₁₀ and PM_{2.5} emissions from stationary sources. In US EPA: Washington, D.C.: 2010.
9. United States Environmental Protection Agency (US EPA). Method 202-Dry impinger method for determining condensable particulate emissions from stationary sources. In US EPA: Washington, D.C.: 2010.
10. Wu, B., et al. (2018). Effects of Wet Flue Gas Desulfurization and Wet Electrostatic Precipitators on Emission Characteristics of Particulate Matter and Its Ionic Compositions from Four 300 MW Level Ultralow Coal-Fired Power Plants. *Environ. Sci. Technol.* 52, 14015.
11. Liu, W., et al. (2020). Migration and Emission Characteristics of Ammonia/Ammonium through Flue Gas Cleaning Devices in Coal-Fired Power Plants of China. *Environ. Sci. Technol.* 54, 390.
12. Tsai, J.-H., et al. (2007). Chemical constituents in particulate emissions from an integrated iron and steel facility. *J. Hazard. Mater.* 147, 111.
13. Guo, Y., et al. (2017). Chemical profiles of PM emitted from the iron and steel industry in northern China. *Atmos. Environ.* 150, 187.
14. Guo, Z., et al. (2021). Field measurements on emission characteristics, chemical profiles, and emission factors of size-segregated PM from cement plants in China. *Sci. Total Environ.* 818, 151822.